# Integrated Growth Model of Typical Crops in China with Regional Parameters

Fangliang Liu [1,2], Yunhe Liu [2], Lijun Su [1,2,3,*], Wanghai Tao [1,2], Quanjiu Wang [1,2] and Mingjiang Deng [1,2]

1 State Key Laboratory of Eco-Hydraulics in Northwest Arid Region, Xi'an University of Technology, Xi'an 710048, China; xautsoilwater@163.com (F.L.); xatwh@xaut.edu.cn (W.T.); wquanjiu@163.com (Q.W.); xjdmj@163.com (M.D.)
2 Institute of Water Resources and Hydroelectric Engineering, Xi'an University of Technology, Xi'an 710048, China; liuyunhe_lyh@163.com
3 School of Science, Xi'an University of Technology, Xi'an 710054, China
* Correspondence: sljun11@163.com

**Abstract:** The analysis of common properties of growth for crops is the basis for further understanding crop growth in different regions. We used four typical crops of China, winter wheat, summer maize, rice, and cotton, to build an integrated model suitable for simulating the growth of different crops. The rates and characteristics of crop growth were systematically analysed based on semirelative and fully relative logistic models of crop growth, and a comprehensive, fully relative logistic model for the four crops was established. The spatial distributions of the maximum leaf area index ($LAI_{max}$) and maximum dry-matter accumulation ($DMA_{max}$) for the four crops were analysed. The semirelative and fully relative growth models exhibited different characteristics of crop growth. The essential characteristics of growth and the characteristics of the crops at each stage of growth were better represented by the fully relative logistic growth model than by the semirelative model. The comprehensive, fully relative logistic model fitted the growth of all four crops well. $LAI_{max}$ and $DMA_{max}$ varied greatly amongst the four crops and were strongly regionally distributed. These indicators for the same crop were differentially spatially variable, and the two indicators were not significantly correlated, except for rice. $LAI_{max}$ and $DMA_{max}$ in different regions could be obtained using a binary quadratic equation of water consumption and growing degree days for the crops. This study provides a novel method for quantitatively judging the status of crop growth, predicting crop yields, and planning for regional agricultural planting.

**Keywords:** crop growth indices; growing degree days; logistic model; spatial variability; winter wheat; maize; rice; cotton

## 1. Introduction

The United Nations Intergovernmental Panel on Climate Change reported that the global average temperature of the terrestrial surface was 0.78 °C higher in 2003–2012 than 1850–1900 and that the atmospheric temperature was expected to increase by 4.8 °C by 2100 (IPCC, 2013). Annual solar radiation, average temperature, and annual active accumulated temperature in China have all tended to increase, despite fluctuations in climate change [1–5]. These factors change the dates for planting crops, the length of the growing season, and appropriate systems of planting management [6]. Climate change is gradually becoming extreme. Rainfall in arid areas is continuing to decrease, and rainfall in humid areas is gradually increasing [7]. Crop growth is sensitive to climate change. Winter wheat, summer maize, and rice are the three main food crops around the world, and cotton is an important cash crop [8–10]. The growth and yields of these four crops have been negatively affected [11–13]. Models of crop growth, as important tools for clarifying the relationship between meteorology and crops and for understanding the mechanisms of crop growth, have attracted considerable attention [4–16].

The crop models commonly used around the world include computer models such as Aquacrop, APSIM, GERES, EPIC, DSSAT, and CropSyst; and mathematical models such as the Gompertz [17], logistic [18] and Richard [19] models. The mathematical models have simpler forms and fewer parameters than the computer models and so are easier to apply [20]. Jiang et al. used the two important factors, temperature and soil water content, to modify several mathematical models, and then used the models to simulate the increase in height of winter-wheat plants. The results indicated that the modified logistic model provided the best fit [21]. Ding et al. used the logistic model to simulate the increase in plant height and accumulation of aboveground dry matter in a winter-wheat/summer-maize rotation system with plastic mulching [22]. Fang et al. used the logistic model to fit the change of 100-kernel weight in summer maize under different treatments of mulching and nitrogen fertilisation [23]. Liu et al. established models for cotton leaves, petioles, and internodes based on a logistic function, and the model fitted the results well [24]. A single model of crop growth has gradually emerged, but a comprehensive comparison and analysis of the characteristics of growth of different crops simulated by the same model are lacking. Liu et al. used the logistic model to describe the growth characteristics of winter wheat in China [25], and Su et al. established a universal growth model for Chinese rice using a logistic function [26]. We integrated the results reported by Su et al. [26] and Liu et al. [25] and further studied the growth of summer maize and cotton in China. We also compared the growth characteristics of winter wheat, summer maize, rice, and cotton to establish a unified comprehensive growth model based on the logistic growth model.

Studies of the spatial variability of regional soil quality [27] and meteorological factors [28] have greatly developed with the extensive application of "3S" technology in agriculture [29], as have predictions of crop yield [30,31]. The maximum values of indices of crop growth in the logistic model greatly influence the final fitted result. The leaf area index (LAI) and dry-matter accumulation (DMA) are two important indicators and are correlated with crop yield [32,33]. LAI can characterise the distribution of crop nutrients during growth. An LAI that is too large or too small is not conducive to the formation of crop yield. DMA can also represent the growth of crops. Studying the distribution of maximum LAI ($LAI_{max}$) and maximum DMA ($DMA_{max}$) is therefore important for predicting crop yields in different regions [34,35]. We analysed the spatial variability of the growth indicators for the four typical crops in China and clarified the relationships between meteorological factors and the indicators to provide a theoretical basis and guidance for planning regional agriculture, developing crop planting systems, and predicting crop yield.

The objectives of this study were to (1) compare the characteristics of growth of four crops based on the logistic model, (2) establish an integrated logistic model for the crops, and (3) clarify the characteristics of spatial distribution and hydrothermal coupling for the indices of maximum crop growth in the logistic model.

## 2. Materials and Methods

### 2.1. Study Sites

Winter wheat, summer maize, rice, and cotton are planted in various regions of China depending on geographic and climatic conditions. Winter wheat is mainly planted in the valleys of the Yellow and Huai Rivers, the middle and lower reaches of the Yangtze River, and the Xinjiang Uygur Autonomous Region (Xinjiang) [36]. Summer maize is mainly planted on the Northern China Plain and in the valleys of the Yellow and Huai Rivers [37]. Rice is mainly planted in northeastern China, the valleys of the Yellow and Huai Rivers, the middle and lower reaches of the Yangtze River, southwestern China, and southern China [38]. Cotton is mainly planted in Xinjiang [39]. Figure 1 shows a map of the planting sites in China for the four crops.

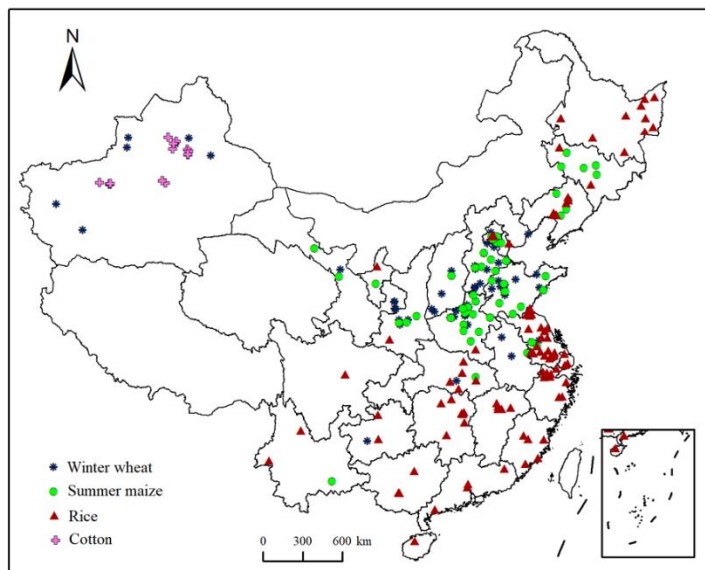

**Figure 1.** Map of the planting sites of the four crops in China.

*2.2. Data Sources*

A meteorological dataset was obtained from the National Meteorological Information Center (http://data.cma.cn/, accessed on 1 January 2018) and included the daily minimum and maximum temperatures and average annual rainfall. The quality of this meteorological dataset is controlled by the China Meteorological Data Service Center (CMDC). Crop data were derived from publications. Liu et al. [25] and Su et al. [26] were the sources of data for winter wheat and rice. We collected data and performed similar treatments on summer maize and cotton as those described by Liu et al. [25]. The main soil types were defined by the International Soil Classification System. Specific information for each site of the four crops is presented in Tables 1–8.

**Table 1.** Geographical Position and Main Characteristics of Each Site of Winter Wheat in China.

| City | Geographical Position | Altitude (m) | Average Annual Rainfall (mm) | Average Annual Temperature (°C) | Main Soil Type |
|---|---|---|---|---|---|
| Handan | 36.63° N, 114.54° E | 55.00 | 553.20 | 13.50 | Sandy loam |
| Shijiazhuang | 38.04° N, 114.51° E | 450.00 | 632.40 | 14.20 | Loam |
| Hengshui | 37.74° N, 115.67° E | 27.40 | 571.02 | 14.51 | Loam |
| Langfang | 39.54° N, 116.68° E | 13.00 | 517.39 | 8.98 | Sandy loam |
| Baoding | 39.02° N, 116.08° E | 16.80 | 500.73 | 12.88 | Sandy loam |
| Qinhuangdao | 39.94° N, 119.60° E | 570.90 | 665.56 | 10.39 | Sandy loam |
| Yucheng | 36.94° N, 116.64° E | 23.40 | 573.92 | 14.04 | Sandy loam |
| Laizhou | 37.18° N, 119.94° E | 25.00 | 687.76 | 12.79 | Loam |
| Jinan | 36.65° N, 117.12° E | 170.30 | 704.01 | 14.90 | Loam |
| Jiaozhou | 36.27° N, 120.03° E | 20.00 | 644.80 | 14.49 | Loam |
| Zibo | 36.81° N, 118.05° E | 60.00 | 586.01 | 13.74 | Sandy loam |
| Liaocheng | 36.46° N, 115.99° E | 23.40 | 600.76 | 14.42 | Clay |
| Beijing | 39.91° N, 116.41° E | 31.30 | 636.30 | 11.47 | Loam |
| Luopu | 37.08° N, 80.20° E | 1356.00 | 163.33 | 4.26 | Sand |
| Zepu | 38.15° N, 77.17° E | 1279.00 | 147.40 | 9.19 | Sandy loam |
| Manasi | 43.92° N, 86.07° E | 608.00 | 306.24 | 6.40 | Sandy loam |
| Wujiaqu | 44.17° N, 87.54° E | 462.00 | 127.18 | 8.70 | Loam |
| Hefei | 31.82° N, 117.23° E | 49.80 | 1111.30 | 16.72 | Clay |
| Gaoyou | 32.78° N, 119.46° E | 6.50 | 1103.10 | 16.24 | Clay |
| Nanjing | 32.06° N, 118.80° E | 35.20 | 1277 | 16.50 | Clay |
| Changshu | 31.66° N, 120.75° E | 11.00 | 945.60 | 15.06 | Clay |
| Hangzhou | 30.25° N, 120.21° E | 41.70 | 1620.00 | 17.70 | Clay |

**Table 1.** *Cont.*

| City | Geographical Position | Altitude (m) | Average Annual Rainfall (mm) | Average Annual Temperature (°C) | Main Soil Type |
|------|----------------------|--------------|------------------------------|--------------------------------|----------------|
| Anshun | 26.25° N, 105.93° E | 143.11 | 1128.50 | 14.48 | Clayey loam |
| Qianjiang | 30.42° N, 112.90° E | 30.80 | 1159.10 | 16.97 | Clay |
| Yangling | 34.23° N, 108.09° E | 521.00 | 610.59 | 11.16 | Loam |
| Xianyang | 34.33° N, 108.71° E | 518.00 | 621.53 | 12.59 | Loam |
| Yuncheng | 35.02° N, 111.00° E | 375.00 | 500.79 | 14.54 | Loam |
| Linfen | 36.08° N, 111.52° E | 449.50 | 479.99 | 14.22 | Loam |
| Jinzhong | 37.70° N, 112.74° E | 902.00 | 549.96 | 12.62 | Loam |
| Zhengzhou | 34.76° N, 113.67° E | 110.40 | 618.91 | 15.96 | Loam |
| Xuchang | 34.02° N, 113.83° E | 67.20 | 682.6 | 14.87 | Loam |
| Luoyang | 34.66° N, 112.43° E | 304.00 | 556.13 | 15.64 | Loam |
| Hebi | 35.75° N, 114.30° E | 102.00 | 593.23 | 14.45 | Loam |
| Anyang | 36.10° N, 114.35° E | 75.50 | 590.25 | 14.42 | Clayey loam |
| Shangqiu | 34.44° N, 115.65° E | 52.00 | 755.01 | 14.70 | Loam |
| Yanshi | 34.73° N, 112.79° E | 184.00 | 587.62 | 14.85 | Loam |
| Lanzhou | 36.06° N, 103.83° E | 151.72 | 127.26 | 8.56 | Sandy loam |
| Cangzhou | 38.31° N, 116.84° E | 8.20 | 517.28 | 13.51 | Loam |
| Yining | 43.98° N, 81.53° E | 813.00 | 236.29 | 4.46 | Clay |
| Qianxian | 34.52° N, 108.25° E | 580.00 | 720.45 | 14.27 | Loam |
| Xinxiang | 35.30° N, 113.88° E | 81.00 | 594.18 | 15.46 | Sandy loam |
| Taiyuan | 37.87° N, 112.55° E | 776.30 | 477.23 | 11.13 | Loam |

**Table 2.** Varieties and Main Treatments at Each Site of Winter Wheat in China. N, N fertilisation (kg·hm$^{-2}$); P, P fertilisation (kg·hm$^{-2}$); K, K fertilisation (kg·hm$^{-2}$); I, irrigation (mm); M, modelling; V, validation.

| City | Varieties | Main Treatments | Used for |
|------|-----------|-----------------|----------|
| Handan | Han 6172, Hanmai 13, Han 4564, Jiaozhuang 3475 | N (375, 450), K 187.5 | M |
| Shijiazhuang | Kenong 199 | N 262.5, P 138 | M |
| Hengshui | Baofeng 104 | N 300, P 175, K 175 | M |
| Langfang | Baofeng 104, Beinong 9549 | N (0, 60, 120, 180, 240, 300), P 75, K 75 | M |
| Baoding | Lukenmai 9, Henong 822 | N 284, P 102, K 95 | M |
| Qinhuangdao | Chaoyou 66 | I (60, 120, 180, 240), N 300 | M |
| Yucheng | Keyu 13 | N 245.3 | M |
| Laizhou | PH 99–31, BY 8175 | N 270, P 135, K 110 | M |
| Jinan | Yannong 19, Jimai 20, Jimai 19, Taishan 23 | N 225, P 450 | M |
| Jiaozhou | Qingmai 7 | I (30, 60, 90, 120, 150, 240), N 108, P (108, 48) | M |
| Zibo | Lumai 103 | I (165, 210), N (0, 100, 200, 300) | M |
| Liaocheng | LN05–1, LN05–2, LN05–3, LN06–1, LN06–2, LN07–1, LN07–2, LN07–3, LN07–4, LN07–5 | N 488.9, P 400, K 333.3 | M |
| Beijing | Zhongyou 9507, Jingdong 8 | I (60, 120), N (0, 75, 150, 225, 300, 375) | M |
| Luopu | Nongda 212, Baomai 10, Henong 825, Henongpin 50, Henong 827, Jingdong 8, 5480, Pin 2, Xingmai 4, Shimai 15, Jimai 22, Guan 35, Xindong 20 | N 525, P 450, K 375 | M |
| Zepu | Xindong 40 | I 360, N 150, P 375 | M |
| Manasi | Xindong 18 | I (232.5, 255, 300, 345, 367.5, 435, 525), N (90, 180, 270, 360, 495) | M |
| Wujiaqu | Xindong 8 | I (270, 360, 450), N (160, 450, 750) | M |
| Hefei | Wanmai 38 | N (120, 240, 360), P 900, K 112.5 | M |
| Gaoyou | Ningmai 9 | N 225, P 300, K 300 | M |
| Nanjing | Ningmai 9, Ningmai 13, Yumai 34 | N (0, 75, 90, 150, 180, 225, 270, 300), P 80, K 150 | M |
| Changshu | Yangmai 10 | N (0, 93.75, 168.75, 243.75), P 56.25, K 56.25 | M |
| Hangzhou | Ningmai 13 | N 275, P 140, K 120 | M |
| Anshun | Anmai 6 | N 225, P 120, K 90 | M |
| Qianjiang | Zhengmai 9023, Wanmai 369 | N 350 | M |

**Table 2.** *Cont.*

| City | Varieties | Main Treatments | Used for |
|---|---|---|---|
| Yangling | Xiaoyan 22 | I (75, 120, 135, 150), N 256.5, P 240 | M |
| Xianyang | Changwu 134 | N (90, 180), P (90, 180) | M |
| Yuncheng | Liangxing 99, Jinmai 79 | I (80, 100, 120), N (150, 200, 250), P 240 | M |
| Linfen | Yaomai 16 | I (240, 320), N (280, 387), P (50, 79), K (108, 120) | M |
| Jinzhong | Ji 22, Lumai 14, Jingdong 8, Jing 9428, Beinongbai, Shannong 9–1, Shannong 9801 | N (75, 150, 225, 300), P 150, K 150 | M |
| Zhengzhou | Yumai 49–198, Yumai 13 | N 180, P 90, K 180 | M |
| Xuchang | Zhoumai 27, Zhengmai 366, Aikang 58, Yumai 49–198 | N (120, 225, 330), P 134.9, K 104.9 | M |
| Luoyang | Yumai 49–198, Yanzhan 4110, Yanshi 918–58 | I (90, 135, 150, 180, 225, 300), N (105, 210, 315), P (37.5, 112.5, 187.5), K 67.5 | M |
| Hebi | Xinmai 26, Yumai 49–198, Bainong 66 | I (135, 142.5, 172.5), N (150, 175, 240), P (112.5, 412.5), K (60, 150) | M |
| Anyang | Zhoumai 16, Zhou 18 | N (100, 180, 200, 225, 270, 300), P (120, 134.9), K (75, 104.9) | M |
| Shangqiu | Yujiao 5 | N (120, 240, 300, 360) | M |
| Yanshi | Yumai 18 | K (75, 150, 225) | M |
| Lanzhou | Shidong 8, Ningdong 6, Jimai 22 | K (150, 195, 240) | M |
| Cangzhou | 9402 | N 450, P 300 | V |
| Yining | Yinong 21 | I (345, 375, 435), N (0, 104, 173, 242), P 300 | V |
| Qianxian | Shan 229 | N (0, 112.5, 187.5, 262.5, 337.5), P (90, 150, 210, 270) | V |
| Xinxiang | Linong 9968 | I 75 | V |
| Taiyuan | Yaomai 16 | N 630, P 345, K 75 | V |

**Table 3.** Geographical Position and Main Characteristics of Each Site for Summer Maize in China.

| City | Geographical Position | Altitude (m) | Average Annual Rainfall (mm) | Average Annual Temperature (°C) | Main Soil Type |
|---|---|---|---|---|---|
| Haicheng | 40.88° N, 122.68° E | 34.40 | 721.30 | 10.40 | Loam |
| Fuxin | 42.02° N, 121.67° E | 153.20 | 565.60 | 8.60 | Loam |
| Tongliao | 44.13° N, 123.31° E | 178.70 | 375.50 | 7.56 | Silty loam |
| Huadian | 43.23° N, 126.51° E | 263.30 | 824.76 | 4.84 | Clayey loam |
| Changchun | 43.81° N, 125.41° E | 236.80 | 649.10 | 6.28 | Loam |
| Jinan | 36.71° N, 117.08° E | 170.30 | 704.01 | 14.90 | Sandy loam |
| Taian | 36.18° N, 117.04° E | 1533.70 | 637.02 | 13.97 | Sandy loam |
| Tengzhou | 35.11° N, 117.17° E | 40.90 | 627.39 | 12.82 | Loam |
| Gunzhou | 35.42° N, 116.59° E | 45.60 | 580.30 | 14.10 | Loam |
| Qingdao | 36.07° N, 120.38° E | 76.00 | 624.74 | 13.33 | Loam |
| Jining | 35.41° N, 116.59° E | 43.70 | 708.50 | 13.70 | Loam |
| Linyi | 35.11° N, 118.36° E | 107.40 | 813.77 | 14.37 | Loam |
| Laiyang | 36.98° N, 120.71° E | 66.30 | 607.19 | 12.58 | Loam |
| Dezhou | 37.44° N, 116.36° E | 27.40 | 600.76 | 14.43 | Loam |
| Laizhou | 37.18° N, 119.94° E | 25.00 | 687.76 | 12.79 | Loam |
| Jiaozuo | 35.22° N, 113.24° E | 113.20 | 699.80 | 13.80 | Clayey loam |
| Pingdingshan | 33.77° N, 113.19° E | 197.20 | 949.50 | 13.55 | Loam |
| Zhumadian | 33.01° N, 114.02° E | 106.20 | 855.28 | 15.60 | Clayey loam |
| Kaifeng | 34.80° N, 114.31° E | 73.70 | 556.13 | 15.61 | Loam |
| Yuzhou | 34.14° N, 113.49° E | 136.60 | 650.00 | 14.50 | Loam |
| Xinxiang | 35.31° N, 113.93° E | 81.00 | 594.18 | 15.46 | Sandy loam |
| Zhengzhou | 34.75° N, 113.63° E | 110.40 | 618.91 | 15.96 | Sandy loam |
| Wenxian | 34.95° N, 113.09° E | 112.00 | 625.00 | 14.50 | Loam |
| Hebi | 35.68° N, 114.56° E | 102.00 | 593.23 | 14.45 | Clayey loam |
| Anyang | 36.11° N, 114.40° E | 75.50 | 590.25 | 14.42 | Loam |
| Shangqiu | 34.42° N, 115.66° E | 52.00 | 755.01 | 14.70 | Loam |
| Zhongmou | 34.73° N, 113.98° E | 108.00 | 616.00 | 14.20 | Sandy loam |
| Luoyang | 34.63° N, 112.46° E | 304.00 | 556.13 | 15.64 | Loam |

**Table 3.** *Cont.*

| City | Geographical Position | Altitude (m) | Average Annual Rainfall (mm) | Average Annual Temperature (°C) | Main Soil Type |
|---|---|---|---|---|---|
| Beijing | 39.91° N, 116.41° E | 31.30 | 636.30 | 11.47 | Sandy loam |
| Quzhou | 36.77° N, 114.96° E | 37.20 | 556.20 | 13.10 | Loam |
| Cangzhou | 38.31° N, 116.85° E | 8.20 | 517.28 | 13.51 | Loam |
| Xinji | 37.95° N, 115.22° E | 37.50 | 586.93 | 12.50 | Sandy loam |
| Shijiazhuang | 38.05° N, 114.52° E | 450.00 | 632.40 | 14.20 | Loam |
| Langfang | 39.54° N, 116.69° E | 13.00 | 517.39 | 8.98 | Sandy loam |
| Baoding | 38.88° N, 115.47° E | 16.80 | 500.73 | 12.88 | Loam |
| Tianjin | 39.09° N, 117.21° E | 3.50 | 523.49 | 13.27 | Loam |
| Wuwei | 37.94° N, 102.64° E | 1540.20 | 178.48 | 9.55 | Silty loam |
| Zhangye | 38.93° N, 100.46° E | 1461.10 | 127.26 | 8.56 | Loam |
| Tongxin | 36.99° N, 105.92° E | 1336.40 | 239.89 | 10.11 | Clayey loam |
| Yangling | 34.30° N, 108.07° E | 521.00 | 610.59 | 11.16 | Loam |
| Weinan | 34.51° N, 109.52° E | 437.40 | 491.45 | 14.15 | Loam |
| Yangzhou | 32.40° N, 119.42° E | 7.30 | 951.30 | 15.65 | Clay |
| Nanjing | 32.07° N, 118.80° E | 35.20 | 1277.00 | 16.50 | Loamy clay |
| Tianchang | 32.67° N, 119.01° E | 43.90 | 917.42 | 14.80 | Clay |
| Fuyang | 32.90° N, 115.82° E | 32.70 | 883.98 | 15.61 | Loamy clay |
| Xianyang | 34.34° N, 108.72° E | 518.00 | 621.53 | 12.59 | Loam |
| Huanghua | 38.38° N, 117.34° E | 5.00 | 580.22 | 13.38 | Sandy loam |
| Xuchang | 34.04° N, 113.86° E | 67.20 | 682.60 | 14.87 | Loam |
| Zaozhuang | 34.81° N, 117.32° E | 114.30 | 603.59 | 13.41 | Sandy loam |
| Jilin | 43.88° N, 126.57° E | 229.50 | 749.80 | 6.02 | Sandy loam |

**Table 4.** Varieties and Main Treatments at Each Site for Summer Maize in China. N, N fertilisation (kg·hm$^{-2}$); P, P fertilisation (kg·hm$^{-2}$); K, K fertilisation (kg·hm$^{-2}$); I, irrigation (mm); M, modelling; V, validation.

| City | Varieties | Main Treatments | Used For |
|---|---|---|---|
| Haicheng | Zhengdan 958 | N 250, P 150, K 180 | M |
| Fuxin | Danyu 39 | N 200, P 187.5 | M |
| Tongliao | Zhengdan 958 | I 280, N 524.25, P 108.75 | M |
| Huadian | Jidan 631 | N 225, P 90, K 120 | M |
| Changchun | Xianyu 335, Zhengdan 958, Sanbei 9, Changcheng 799, Tongdan 258, Huake 425, Donghua 106, Yuyu 22, Nongda 518 | N (280, 330), P (100, 180), K (60, 100) | M |
| Jinan | Nuoda 1 | | M |
| Taian | Denghai 661, Zhengdan 958, Yedan 22 | N (160.5, 184.5, 225, 450), P (45, 55.5, 75, 150), K (75, 130.5, 150, 300) | M |
| Tengzhou | Denghai 661, Zhengdan 958, Nongda 108 | N 300, P 120, K 240 | M |
| Gunzhou | Denghai 661, Zhengdan 958, Nongda 108 | N 300, P 120, K 240 | M |
| Qingdao | Qingnong 8 | I (90, 180, 270, 360), N (150, 210, 270, 330), P (60, 120, 180, 240) | M |
| Jining | Zhengdan 958 | N (150, 225) | M |
| Linyi | Tiantai 33, Tiantai 55, Zhendgan 958 | N 360, P 150, K 300 | M |
| Laiyang | Nongda 106, Yedan 22 | N 900, P 75 | M |
| Dezhou | Zhengdan 958, Denghai 618 | N 305 | M |
| Laizhou | Jinhai 5 | N 225, P 135, K 180 | M |
| Jiaozuo | Yedan 13, Yedan 22, Zhengdan 958, Denghai 601 | N (121.5, 300, 478.5, 600, 750), P (45, 112.5, 180, 225, 450), K (76.5, 187.5, 298.5, 375, 600) | M |
| Pingdingshan | Xundan 18 | N 260, P 125, K 100 | M |
| Zhumadian | Chuangyu 198, Yuyu 2, Yuyu 5, Zhengdan 958, Xundan 22 | N (135, 138, 225), P (45, 48, 75, 150), K (75, 135, 150) | M |
| Kaifeng | Zhengdan 958 | | M |
| Yuzhou | Zhengdan 958, Xundan 20 | N (86.25, 350), P (114, 150), K (81, 180) | M |
| Xinxiang | Zhengdan 958, Xundan 18, Xundan 20, Xindan 2 | I (90, 120), N (350, 506.25), P 150, K 180 | M |
| Zhengzhou | Xundan 20, Yedan 22 | I 52.5, N (350, 750), P (56.25, 150, 450), K (56.25, 150, 600) | M |
| Wenxian | Zhengdan 17, Yedan 22, Zhendgan 958, Xundan 20 | N (600, 1200), P (600, 1200) | M |
| Hebi | Zhengdan 958, Xundan 20, Xundan 22, Denghai 3719 | N (270, 345, 450), P (90, 225), K (25, 120, 225) | M |
| Anyang | Zhengdan 958, Yuyu 25, Denghai 661 | N (225, 450, 675), P (150, 300, 450), K 450 | M |
| Shangqiu | Jixiang 1, Qiaoyu 8, Zhengdan 958 | N (530, 750), P (205, 1200), K (450, 540) | M |
| Zhongmou | Yedan 22 | N 489, P 804 | M |
| Luoyang | Luoyu 8, Yuyu 28, Luoyu 863, Luoyu 818 | N (390, 585, 600), P (135, 900), K (168.5, 255, 303.5) | M |
| Beijing | Jingken 114, Jiyuan 101 | I (83.3, 86.9, 91.1, 98.9, 99, 99.79, 100.1, 100.4, 103.2, 108.1, 111.8, 114.2), N (12.45, 112.45, 212.45, 300), P (12.45, 126), K 12.45 | M |
| Quzhou | Nongda 108 | I 157.5 | M |

**Table 4.** *Cont.*

| City | Varieties | Main Treatments | Used For |
|---|---|---|---|
| Cangzhou | CF008, Zhengdan 958, Jinhai 5 | N (90, 180, 210, 270), P (90, 103.5), K (60, 112.5) | M |
| Xinji | Zhengdan 958 | I (180, 250, 350, 450, 520), N (75, 180, 290, 365), K (50, 120, 190, 235) | M |
| Shijiazhuang | Yongyu 1, Xianyu 335, Zhengdan 958 | N 250, P 125 | M |
| Langfang | Yedan 4, Yedan 12, Yedan 13, Danyu 13, Yinong 103, Xianyu 335, Denghai 661 | N (80, 363), P (30, 172.5), K 150 | M |
| Baoding | Yedan13 | N (135, 270, 540), P (90, 180, 360) | M |
| Tianjin | Jiyuan 1 | N (225, 300, 375), P (90, 112.5, 120, 150, 187.5), K (90, 120, 135, 150, 180, 225) | M |
| Wuwei | Funong 963 | I (225, 325, 450), N 262.5, P 525 | M |
| Zhangye | Longdan 3 | N 300, P 270 | M |
| Tongxin | Xianyu 335 | I (135, 165, 210, 255), N 204, P 68, K 54.4 | M |
| Yangling | Qinlong 11, Shandan 10, Shandan 8806, Zhengdan 958 | I30, N (225, 450), P (90, 225), K 450 | M |
| Weinan | Xundan 29 | N 300, P 120 | M |
| Yangzhou | Suyu 31, Suyu 33 | N 300, P 120, K 150 | M |
| Nanjing | Jiangyu 403 | N (75, 112.5) | M |
| Tianchang | Denghai 11 | N 225, P 75, K 150 | M |
| Fuyang | Anlong 4, Ludan 981, Zhengdan 958, Liyu 16, Zhongke 11, Huadan 986 | N 345, P 67.5, K 67.5 | M |
| Xianyang | Yudan 6 | N 300, P 200 | V |
| Huanghua | Zhengdan 958 | N 300 | V |
| Xuchang | Zhengdan 958 | N 75, P 105, K 75 | V |
| Zaozhuang | Huawan 602, Denghai 605, Denghai 618, Longping 206, Longping 208, Qidan 1, Zhengdan 958 | N 300 | V |
| Jilin | Zhengdan 958 | N (90, 180, 270) | V |

**Table 5.** Geographical Position and Main Characteristics of Each Site for Rice in China.

| City | Geographical Position | Altitude (m) | Average Annual Rainfall (mm) | Average Annual Temperature (°C) | Main Soil Type |
|---|---|---|---|---|---|
| Changsha | 28.23° N, 112.94° E | 42.00 | 1447.90 | 16.50 | Clayey loam |
| Qiyang | 26.58° N, 111.84° E | 172.60 | 1410.41 | 18.52 | Clayey loam |
| Yueyang | 29.36° N, 113.13° E | 53.00 | 1396.24 | 17.95 | Clay |
| Liling | 27.65° N, 113.50° E | 114.00 | 1450.00 | 18.00 | Clay |
| Yiyang | 28.56° N, 112.36° E | 102.00 | 1465.00 | 16.50 | Loam |
| Dawa | 41.00° N, 122.08° E | 3.00 | 645.00 | 8.30 | Clayey loam |
| Panjin | 41.12° N, 122.07° E | 3.30 | 446.60 | 9.90 | Sandy loam |
| Shenyang | 41.68° N, 123.46° E | 51.00 | 681.06 | 8.30 | Clayey loam |
| Jingshan | 31.02° N, 113.12° E | 77.00 | 1179.00 | 16.30 | Clayey loam |
| Wuhan | 30.59° N, 114.31° E | 23.60 | 1322.61 | 16.97 | Sandy loam |
| Suizhou | 31.69° N, 113.38° E | 122.00 | 967.50 | 15.50 | Clay |
| Wuxi | 31.49° N, 120.31° E | 5.30 | 1121.70 | 16.20 | Clayey loam |
| Yancheng | 33.35° N, 120.16° E | 2.50 | 882.47 | 14.13 | Sandy loam |
| Changshu | 31.66° N, 120.75° E | 4.10 | 1615.30 | 16.90 | Sandy loam |
| Lianyungang | 34.60° N, 119.22° E | 4.70 | 883.60 | 14.00 | Clayey loam |
| Nanjing | 32.06° N, 118.80° E | 35.20 | 1277.00 | 16.50 | Clayey loam |
| Huaian | 33.61° N, 119.02° E | 12.50 | 945.60 | 15.06 | Clay |
| Changzhou | 31.81° N, 119.97° E | 7.60 | 1149.70 | 17.50 | Clayey loam |
| Zhangjiagang | 31.88° N, 120.56° E | 5.40 | 957.04 | 14.81 | Sandy loam |
| Hangzhou | 30.28° N, 120.16° E | 41.70 | 1620.00 | 17.70 | Clayey loam |
| Ningbo | 29.88° N, 121.55° E | 9.40 | 1480.00 | 16.40 | Clay |
| Huzhou | 30.89° N, 120.09° E | 194.30 | 1270.50 | 14.75 | Clay |
| Jiaxing | 30.75° N, 120.76° E | 7.30 | 1168.60 | 15.90 | Clayey loam |
| Cixi | 30.17° N, 121.27° E | 5.40 | 1561.71 | 17.75 | Loamy clay |
| Linhai | 28.86° N, 121.14° E | 302.10 | 1424.94 | 17.10 | Clayey loam |
| Anji | 30.64° N, 119.68° E | 247.40 | 1861.40 | 17.00 | Loam |
| Yujiang | 28.21° N, 116.82° E | 33.20 | 1758.00 | 17.60 | Clay |
| Nanchang | 28.68° N, 115.86° E | 47.20 | 1751.92 | 18.76 | Clayey loam |

**Table 5.** *Cont.*

| City | Geographical Position | Altitude (m) | Average Annual Rainfall (mm) | Average Annual Temperature (°C) | Main Soil Type |
|---|---|---|---|---|---|
| Wenjiang | 30.68° N, 103.86° E | 547.70 | 936.12 | 16.41 | Clayey loam |
| Meixian | 24.29° N, 116.12° E | 116.00 | 1551.09 | 21.83 | Clay |
| Guangzhou | 23.13° N, 113.26° E | 70.70 | 2119.89 | 22.03 | Clay |
| Sanming | 26.40° N, 117.79° E | 285.00 | 1700.00 | 18.20 | Clay |
| Zhangzhou | 24.51° N, 117.65° E | 205.00 | 1860.89 | 19.30 | Clayey loam |
| Harbin | 45.80° N, 126.54° E | 118.30 | 541.62 | 4.94 | Loam |
| Hulin | 45.76° N, 132.94° E | 98.10 | 614.81 | 3.95 | Loam |
| Mudanjiang | 44.55° N, 129.63° E | 305.70 | 587.03 | 4.62 | Clay |
| Fujin | 47.25° N, 132.04° E | 66.40 | 550.04 | 3.19 | Loam |
| Kiamusze | 46.80° N, 130.32° E | 82.00 | 638.89 | 3.72 | Sandy loam |
| Daan | 45.51° N, 124.29° E | 132.10 | 413.70 | 4.30 | Clayey loam |
| Tonghua | 41.73° N, 125.94° E | 402.90 | 891.99 | 6.20 | Loam |
| Hanzhong | 33.16° N, 107.33° E | 509.50 | 908.21 | 15.66 | Loam |
| Zunyi | 27.73° N, 106.93° E | 753.30 | 930.87 | 15.10 | Sandy loam |
| Guiyang | 26.66° N, 106.63° E | 1227.30 | 1102.46 | 14.75 | Sandy loam |
| Liuzhou | 24.33° N, 109.42° E | 306.00 | 1479.10 | 21.26 | Sandy loam |
| Shanghai | 31.23° N, 121.47° E | 5.50 | 1294.11 | 17.30 | Loam |
| Qingtongxia | 38.02° N, 106.08° E | 1131.00 | 260.70 | 8.50 | Loam |
| Nanning | 22.82° N, 108.37° E | 152.00 | 1311.33 | 21.75 | Clay |
| Qiqihar | 47.36° N, 123.92° E | 146.70 | 462.37 | 4.33 | Sandy loam |
| Yangzhou | 32.39° N, 119.41° E | 7.30 | 951.30 | 15.65 | Sandy loam |
| Jingzhou | 30.34° N, 112.24° E | 31.80 | 1071.90 | 17.13 | Clayey loam |
| Beijing | 39.91° N, 116.41° E | 31.30 | 636.30 | 11.47 | Loam |

**Table 6.** Varieties and Main Treatments at Each Site for Rice in China. N, N fertilisation (kg·hm$^{-2}$); P, P fertilisation (kg·hm$^{-2}$); K, K fertilisation (kg·hm$^{-2}$); I, irrigation (mm); M, modelling; V, validation.

| City | Varieties | Main Treatments | Used For |
|---|---|---|---|
| Changsha | Shanyou 64, Luliangyou 996, Jinyou 402, Zhongjiazao 17, Yueyou 360, Peiai 64S/R292, Y58S/R292, Fengyuanyou 299, Zhuliangyou 90, Yueyou 9113, Xiangzaocan 5 | N (90, 105, 120, 135, 150, 165, 375, 480), P (240, 375), K (112.5, 120) | M |
| Qiyang | Unbongbyeo | N (55, 77, 110), P 45, K 57 | M |
| Yueyang | Xiangzaocan 24, Jinyou 207 | N (93, 176, 180, 177, 226, 230), P (503, 753), K (133, 142, 256) | M |
| Liling | Xiangzaocan 45, Fengyuanyou 299 | N (150, 180), P (60, 75), K (90, 120) | M |
| Yiyang | T You705, Xiangfengyou 103, Jinyou 974, Fengyuanyou 272, Jinyou 402, T You 6135, Xiangzaocan 45, Xiangwancan 12 | N 196.5, P 90, K (90, 117) | M |
| Dawa | Yanfeng 47 | N (135, 187.5, 240, 270, 292.5, 345), P (103.5, 105, 135), K (45, 67.5, 75, 90, 135, 150, 180) | M |
| Panjin | Shennong 265, Yanjing 377, Qiaoke 951, Yanfeng 47, Yanjing 218 | N (135.57, 180, 188.01, 225, 240.45, 270, 292.2, 315, 345.33, 360), P 105, K 52.5, 75 | M |
| Shenyang | Liaojing 294, Liaojing 371, Shennong 265, Liaojing 326, Aoyu 316, Qiuguang, Liaojing 294, Shennong 606, Liaoxing 1, Yanfeng 47, Liaojing 9, Shen 98–20, Liaojing 5, Shendao 4, Fengjin, Nonglin 313, Shennong 91, Liaojing 9 | N (90, 120, 150, 160, 180, 210, 487.5), P (13.05, 26.25, 39.3, 41.4, 52.35, 65.4, 90, 300), K (49.8, 90, 99.6, 149.4, 199.2, 225, 249) | M |
| Jingshan | Shanyou 63, Shanyou 6, Zhongxian 910, 75632 | N (112.5, 157.5, 202.5), P 75, K 60 | M |
| Wuhan | Jiannanbaigu, Shenglixian, Xinteqing, Shanyou 63 | N 112.5, P 600 | M |
| Suizhou | Yangliangyou 6, P88S/747, Luoyou 8, Luoyou 234, Tianliangyou 2 | N (195, 240), P (60, 120), K (60, 330) | M |
| Wuxi | Wuxiangjing 14, Shanyou 63 | N (150, 250, 350), P (35, 70) | M |
| Yancheng | Liangyou 363, Xudao 3, Wuyujing 3, Huaidao 5 | N (77.25, 153, 232.5, 300, 319.65, 345, 375, 439.5, 631.65, 768.45), P 75, K 150 | M |
| Changshu | Changyou 1, Liangyoupeijiu, Youming 86, You 084, D You 527, P88S/0293, Shanyou 63 | N (200, 202.5, 216, 225, 229.5, 240, 270), P (40, 112.5, 174, 187.5), K (70, 118.5, 225, 375) | M |
| Lianyungang | Lianjiajing 2, Huajing 5, 0026, 9823, Lianjing 7 | N (248.4, 265.7, 269.1, 282.9), P 600, K 240 | M |
| Nanjing | Wuyujing 7, Wuyujing 3, Teyou 559, Liangyoupeijiu, Shanyou 63, Takanari, IR72, Sankeiso, CH86, IR65564–44–2-2, Nipponbare, Banten | N (147, 219, 225, 294), P 120, K 120 | M |
| Huaian | Huaidao 11, Yongyou 2640 | N 315 | M |
| Changzhou | Wuyunjing 19 | N 270, P 60, K 135 | M |
| Zhangjiagang | Youjing 5356, Zhongyou 1 | N (76.5, 135, 142.5, 190.5), P (34.5, 45), K (84, 112.5) | M |
| Hangzhou | Xieyou 9308, Xiushui 63, Xiushui 110, Bing 9904, Bing 98110 | N (45, 120, 135, 225, 240, 315) | M |
| Ningbo | Yongyou 12 | N (270, 300, 330), P (900, 1050, 1250), K (600, 675, 750) | M |
| Huzhou | Bing 9904, Yongyou 538, Xiushui 134 | N (70, 140, 210, 280), P 990, K 750 | M |
| Jiaxing | Jiayu 293, Bing 93390, You 161 | N (120, 142.5, 165, 187.5, 210, 232.5, 375, 450), P (300, 375), K (112.5, 150, 180) | M |

**Table 6.** *Cont.*

| City | Varieties | Main Treatments | Used For |
|---|---|---|---|
| Cixi | Shanyou 63 | N (150, 225, 300, 375, 450), P 600, K 225 | M |
| Linhai | Liangyoupeijiu, Jiayou 99 | N (160.5, 189, 207, 229.5) | M |
| Anji | Xieyou 413 | N 180 | M |
| Yujiang | Youhang 2 | N (195, 288), P 72, K 195 | M |
| Nanchang | Youming 86 | N (105, 150, 195, 240, 285) | M |
| Wenjiang | Fuyou 838, Chuanxiang 9838 | N 150, P 500 | M |
| Meixian | Shanyou 63 | N 120, P 300 | M |
| Guangzhou | Yuxiangyouzhan, Peizataifeng, Tengxi 138, Peiai 64s/E32, Peiai 64s/9311, Yueza 122, Tesanai 2, Yuexiangzhan, Guangfenxgiang 8, Hemeizhan, Xiangdao 1 | I (204.82, 267.77, 294.55), N (100, 150, 200, 300, 187.5), P (90, 100, 375), K 150 | M |
| Sanming | Teyou 73, Youhang 1 | N (195, 203.25, 300.15), P (79.2, 125.7, 300), K (90, 225, 255) | M |
| Zhangzhou | Zhangfeng 8, 78130 | N 190, P 170, K 150 | M |
| Harbin | Dongnong 423, Dongnong 425, Songjing 9, Longdao 5, Longjing 14, Tengxi 138, Longdao 3, Hejiang 19 | I (414.2, 484.9, 571.1), N (120, 150, 171.5, 346.9, 514.4, 685.8), P (70, 75, 120), K (37.5, 50, 100) | M |
| Hulin | Zhonglongxiang 1, Longyang 16 | N 200, P 150, K 120 | M |
| Mudanjiang | Mudanjiang 32, Duxiang 1, Longdao 5, Songjing 9, Mudanjiang 19 | N (100, 125, 150, 160, 220), P 50, K 120 | M |
| Fujin | Longjing 46, Kongyu 131 | N 105, P 60, K 75 | M |
| Kiamusze | Kongyu 131, Kenjing 1 | N 390, K (220.8, 330, 552) | M |
| Daan | Changbai 9 | N 298, P 90, K 138 | M |
| Tonghua | Nongda 3 | N 120, P 51.75, K 56.3 | M |
| Hanzhong | Changbai 9 | N 330, P 120, K 75 | M |
| Zunyi | Maoxiang 2, Feiyouduo 1, Gangyou 151 | N (157.5, 211.2), P (90, 123), K 150 | M |
| Guiyang | Yunguang 14, Huailiangyou 527, Q You 6, You 838, Qiannanyou 2058 | N (150, 240), P (90, 120), K (180, 240) | M |
| Liuzhou | Xinfengliangyou 6, Fengliangyou 1, Fuxiangyou 98, Fuyaomei 3, Lingyou 6602 | N 180, P 120, K 120 | M |
| Shanghai | Huayou 14, 9734 | N (225, 300, 375, 525), P 60, K 60 | M |
| Qingtongxia | Jingdao 92 | N 525 | M |
| Nanning | Qixuan 42 | N 37.5 | V |
| Qiqihar | Suijing 4 | N 135, P 46.9, K 60 | V |
| Yangzhou | Shanyou 63, Xianyou 63, IR661, Yangdao 4, Suxiejing 1, Yanjing 2, Wuyujing 3, Guanglingxiangjing, Yangjing 4227, Zhendao 88, Huaidao 5, C Liangyou 608, Y Liangyou 1, Xiangjing 97–3017 | N (157.5, 172.5, 225, 247.5, 292.5, 321.75) | V |
| Jingzhou | Ganxin 203, Fengliangyou 1, Xiangfengyou 9 | N 150, P 100, K 100 | V |
| Beijing | IR75, IR7521 7H, PSBRC52, Mestizo | N (75, 145 215), P 30, K 40 | V |

**Table 7.** Geographical position and main characteristics of each site for cotton in China.

| City | Geographical Position | Altitude (m) | Average Annual Rainfall (mm) | Average Annual Temperature (°C) | Main Soil Type |
|---|---|---|---|---|---|
| Altay | 44.32° N, 86.06° E | 735.30 | 237.06 | 4.75 | Sandy loam |
| Shihezi | 44.30° N, 86.06° E | 412.00 | 198.00 | 11.03 | Sandy loam |
| Korla | 41.58° N, 86.17° E | 892.00 | 75.79 | 12.41 | Sandy loam |
| Aksu | 40.46° N, 80.37° E | 1107.10 | 102.36 | 11.75 | Loam |
| Alar | 40.55° N, 81.28° E | 1012.20 | 62.29 | 10.97 | Loam |
| Changji | 44.15° N, 87.46° E | 600.00 | 181.70 | 13.10 | Loam |
| Yining | 43.91° N, 81.28° E | 646.00 | 245.10 | 10.50 | Loam |

**Table 8.** Varieties and Main Treatments at Each Site for Cotton in China. N, N fertilisation (kg·hm$^{-2}$); P, P fertilisation (kg·hm$^{-2}$); K, K fertilisation (kg·hm$^{-2}$); I, Irrigation (mm); M, modelling; V, validation.

| City | Varieties | Main Treatments | Used For |
|---|---|---|---|
| Altay | Xinluzao 45, Zhongmiansuo 50, Xinluzao 45, 45–21 | I (595.7, 608.7, 699.4, 761.9), N (563, 609, 628, 644), P (169, 201, 221), K (169, 201, 221) | M |
| Shihezi | Xinluzao 48, Xinluzao 51, Xinluzao 42 | I (240, 275, 360, 375, 420, 480, 475, 600), N (150, 300, 450, 600, 900), P (120, 300), K 300 | M |
| Korla | Xinluzhong 26 | I (390, 450), N (300, 450, 600, 750), P 210, K 90 | V |
| Aksu | Xinhai 14, Mianzhongmian 35 | N 252, P 355.5, K 177 | V |
| Alar | Xinhai 14, Xinluzhong 67, Zhongmian 35 | I 360, N 736.5, P 297, K 58.5 | V |
| Changji | T10 | I (300, 375, 450, 525) | V |
| Yining | Xinluzao 33, Lumianyan 24, Xinluzao 60, Biaoza A1, Jinza 9, Xinluzao 31 | I 555 | V |

### 2.3. Logistic Growth Model

Growing degree days (GDD) is an important factor representing the resources of light and heat required for crop growth. We calculated GDD as described by Liu et al. [25]. The biological upper and lower temperature limits for the four crops are presented in Table 9. We used the logistic equation to construct growth models for the four crops, with GDD or relative GDD (RGDD) as independent variables. LAI, plant height (H), and DMA were used as indicators of crop growth, and the semirelative logistic model (Equation (1) and fully relative logistic model (Equation (2)) were used to analyse the growth characteristics.

$$Ry = \frac{1}{1 + e^{a_s - b_s \cdot \text{GDD} + c_s \cdot \text{GDD}^2}} \tag{1}$$

$$Ry = \frac{1}{1 + e^{a_f - b_f \cdot \text{RGDD} + c_f \cdot \text{RGDD}^2}} \tag{2}$$

where $Ry$ is an index of relative crop growth, e.g., relative LAI (RLAI), relative H (RH), or relative DMA (RDMA), calculated by dividing a measured growth index by the maximum growth index throughout growth; GDD is the demand for crop growth to a specific stage (°C); RGDD is relative GDD, calculated by dividing GDD by the theoretical maximum GDD (GDD$_{\text{max}}$) throughout crop growth; and $a_s$, $b_s$, $c_s$, $a_f$, $b_f$, and $c_f$ are parameters. $c_s = c_f = 0$ when $Ry$ is RH or RDMA.

**Table 9.** Biological Upper and Lower Temperature Limits for the Four Crops.

| Index | Winter Wheat | Summer Maize | Rice | Cotton |
|---|---|---|---|---|
| Lower temperature limit (°C) | 0 | 7 | 10 | 10 |
| Upper temperature limit (°C) | 32 | 40 | 40 | 40 |

The period of crop growth in the fully relative logistic growth model (Equation (2)) is the theoretical harvest period when RGDD = 1, and GDD in this period is GDD$_{\text{max}}$. Wang et al. found that $a_f + c_f = b_f$ in the fully relative logistic mode [37]. When RGDD = 1 and $Ry = 0.5$ (Equation (2)), GDD$_{\text{max}}$ can then be calculated when $Ry = 0.5$ in Equation (1):

$$\text{GDD} \frac{-b + \sqrt{b^2 - 4ac}}{2c}_{\text{max}} \tag{3}$$

The first-order derivation of Equation (1) is the relationship between the growth rate and GDD (Equation (4)). Let $c = 0$ in Equation (1), and let $\frac{d^2 Ry}{d\text{GDD}^2} = 0$; GDD$_0$ when the crop grows fastest can then be calculated (Equation (5)). Let GDD = GDD$_0$ in Equation (4); the maximum growth rate, $v_{\text{max}}$, can then be calculated (Equation (6)). Let $c = 0$ in Equation (1) and let $\frac{d^3 Ry}{d\text{GDD}^3} = 0$; GDD$_1$ and GDD$_2$ can then be calculated, which represent the inflection points of crop growth from slow to fast and fast to slow, respectively (Equations (7) and (8)). The difference between GDD$_1$ and GDD$_2$ is the GDD demand during the period of vigorous crop growth.

$$v = \frac{(b_s - 2c_s \cdot \text{GDD})e^{a_s - b_s \cdot \text{GDD} + c_s \cdot \text{GDD}^2}}{\left(1 + e^{a_s - b_s \cdot \text{GDD} + c_s \cdot \text{GDD}^2}\right)^2} \tag{4}$$

$$\text{GDD}_0 = \frac{a_s}{b_s} \tag{5}$$

$$v = v_{\text{max}} = \frac{b_s}{4} \tag{6}$$

$$\text{GDD}_1 = \frac{a_s - \ln(2 + \sqrt{3})}{b_s} \tag{7}$$

$$\text{GDD}_2 = \frac{a_s - \ln(2 - \sqrt{3})}{b_s} \tag{8}$$

where $v$ is the rate of increase in an index (d$^{-1}$), GDD$_0$ is GDD when an index of crop growth increases fastest (°C), $v_{\max}$ is the maximum rate of increase in the index (d$^{-1}$), GDD$_1$ is GDD when crop growth is from slow to fast (°C), and GDD$_2$ is GDD when crop growth is from fast to slow (°C). In the fully relative logistic model, GDD$_0$, GDD$_1$, and GDD$_2$ were replaced with RGDD$_0$, RGDD$_1$, and RGDD$_2$, respectively; and $a_s$, $b_s$, and $c_s$ were replaced with $a_f$, $b_f$, and $c_f$, respectively.

We compared the semirelative and fully relative logistic growth models of the four crops, analysed the parameters of each model to identify the characteristics of crop growth and requirements for light and heat, and established a comprehensive, fully relative logistic growth model. We used the coefficient of determination ($R^2$), root mean square error (RMSE), and relative error (RE) to evaluate the results of model fitting.

*2.4. Calculation of LAI$_{max}$ and DMA$_{max}$*

LAI$_{\max}$ and DMA$_{\max}$ for the crops at each planting site were averaged, and spatial interpolation was performed using inverse distance weighting in ArcMap to compare and analyse their spatial distributions. In addition, we conducted Kolmogorov–Smirnov (K–S) tests for LAI$_{\max}$ and DMA$_{\max}$. The relationships between the maximum index of growth, water consumption throughout growth (W), and GDD$_{\max}$ were established using a binary quadratic coupling equation.

Inverse distance weighting is a deterministic method of interpolation in analyses of spatial interpolation; the smaller the distance between the interpolation and measured points, the more similar their properties. The general equation of inverse distance interpolation is:

$$Z(s_0) = \sum_{i=1}^{N} \lambda_i Z(s_i) \tag{9}$$

where $s_0$ is the interpolation point, $Z(s_0)$ is the interpolation result at $s_0$, $s_i$ is the measured point around $s_0$ ($i = 1, \dots, N$, where $N$ is the number of measured points), $Z(s_i)$ is the measured value at $s_i$, and $\lambda_i$ is the weighted value of $s_i$:

$$\lambda_i = \frac{d_{i0}^{-p}}{\sum_{i=1}^{N} d_{i0}^{-p}} \sum_{i=1}^{N} \lambda_i = 1 \tag{10}$$

where $d_{i0}$ is the distance between $s_0$ and $s_i$, and $p$ is a parameter. We used the default value $p = 2$ in ArcMap.

## 3. Results

### 3.1. Characteristics of the Semirelative Logistic Growth Model for the Four Crops

The fitted parameters of the semirelative logistic growth model for the four crops needed to study the characteristics of the growth indices for different crops using GDD are presented in Table 10. Semirelative logistic growth curves for the indices are shown in Figure 2. RLAI for the four crops tended to first increase and then decrease as GDD increased (Figure 2A). The GDD demand for maximum RLAI was lowest for rice, followed by cotton, winter wheat, and summer maize. The GDD demand throughout growth for the four crops was also in the order rice < cotton < winter wheat < summer maize. RH at the same GDD was largest for rice, followed by cotton, summer maize, and winter wheat (Figure 2B). The GDD demand at the same RH was highest for rice and lowest for winter wheat. The variable trend of RDMA indicated that the rate of accumulation of dry matter differed amongst the crops, and the demand for GDD for the same crop differed between the early and late periods of growth (Figure 2C).

**Table 10.** Parameters of the Semirelative Logistic Model for the Indices of Crop Growth. RLAI-GDD, RH-GDD, and RDMA-GDD are the logistic models between RLAI and GDD, RH and GDD, and RDMA and GDD, respectively. $a_s$, $b_s$, and $c_s$ are parameters of the semirelative logistic growth model. $c_s = 0$ in the relationship between RH-GDD and RDMA-GDD.

| Crop | RLAI-GDD | | | RH-GDD | | RDMA-GDD | |
|---|---|---|---|---|---|---|---|
| | $a_s$ | $b_s$ | $c_s$ | $a_s$ | $b_s$ | $a_s$ | $b_s$ |
| Winter wheat | 15.270 | 0.027 | $1.07 \times 10^{-5}$ | 3.233 | 0.004 | 5.273 | 0.005 |
| Summer maize | 9.136 | 0.016 | $5.83 \times 10^{-6}$ | 3.266 | 0.005 | 3.803 | 0.004 |
| Rice | 5.717 | 0.015 | $7.73 \times 10^{-6}$ | 2.172 | 0.005 | 3.199 | 0.003 |
| Cotton | 9.619 | 0.021 | $8.54 \times 10^{-6}$ | 2.976 | 0.006 | 3.946 | 0.004 |

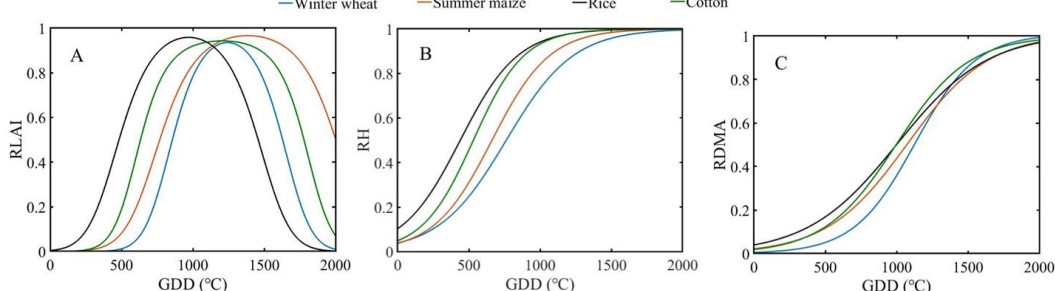

**Figure 2.** Semirelative logistic curves for each crop for (**A**) RLAI, (**B**) RH, and (**C**) RDMA.

We calculated $GDD_0$, $v_{max}$, $GDD_1$, and $GDD_2$ to visually indicate the properties of crop growth (Table 11). The GDD demand needed for the four crops to reach the maximum growth rate ($GDD_0$) was 365–640 °C higher for RDMA than RH. When RH increased at $v_{max}$ for a period of time, RDMA began to change from a slow to a rapid increase, which reached $GDD_1$. When RH changed from a rapid to a slow increase, the dry matter of each crop gradually began to accumulate at the maximum rate (to $GDD_0$). The GDD demand for H during vigorous growth was highest for winter wheat at about 620 °C, lowest for cotton at about 470 °C, and intermediate for rice and summer maize. The GDD demand for DMA for the four crops during vigorous growth was in the order rice > summer maize > cotton > winter wheat, which differed from the order for the GDD demand for increasing H. RDMA for summer maize and cotton during vigorous growth needed more GDD than did RH, and $v_{max}$ was smaller than RH, indicating that these two crops had a higher demand for resources of light and heat in the later period of growth. RH for winter wheat increased slowly in the early stage, and dry matter accumulated rapidly in the later stage, indicating that the GDD demand was opposite for summer maize and cotton.

**Table 11.** Characteristic Values of the Semirelative Logistic Model for the Crops. $GDD_0$ is GDD when RH or RDMA increases the fastest. $v_{max}$ is the maximum rate of increase in RH or RDMA. $GDD_1$ is GDD when RH or RDMA increases from slow to fast. $GDD_2$ is GDD when RH or RDMA increases from fast to slow. $GDD_2 - GDD_1$ is the GDD demand for RH or RDMA during vigorous growth.

| | Crop | $GDD_0$ (°C) | $v_{max}$ (d$^{-1}$) | $GDD_1$ (°C) | $GDD_2$ (°C) | $GDD_2 - GDD_1$ (°C) |
|---|---|---|---|---|---|---|
| RH-GDD | Winter wheat | 762.50 | $1.1 \times 10^{-3}$ | 451.90 | 1073.10 | 621.21 |
| | Summer maize | 661.94 | $1.2 \times 10^{-3}$ | 395.02 | 928.85 | 533.83 |
| | Rice | 367.96 | $1.1 \times 10^{-3}$ | 78.13 | 657.78 | 579.65 |
| | Cotton | 531.43 | $1.4 \times 10^{-3}$ | 296.26 | 766.60 | 470.34 |
| RDMA-GDD | Winter wheat | 1127.91 | $1.2 \times 10^{-3}$ | 846.21 | 1409.62 | 563.40 |
| | Summer maize | 1076.73 | $0.9 \times 10^{-3}$ | 703.86 | 1449.59 | 745.73 |
| | Rice | 1005.03 | $0.8 \times 10^{-3}$ | 591.28 | 1418.77 | 827.49 |
| | Cotton | 996.46 | $1.0 \times 10^{-3}$ | 663.90 | 1329.03 | 665.13 |

The relationships between the growth rates of the crops and GDD are shown in Figure 3. RH and RDMA for the crops tended to first increase and then decrease as GDD increased. The characteristic growth rate for each crop was consistent with the results in Table 10.

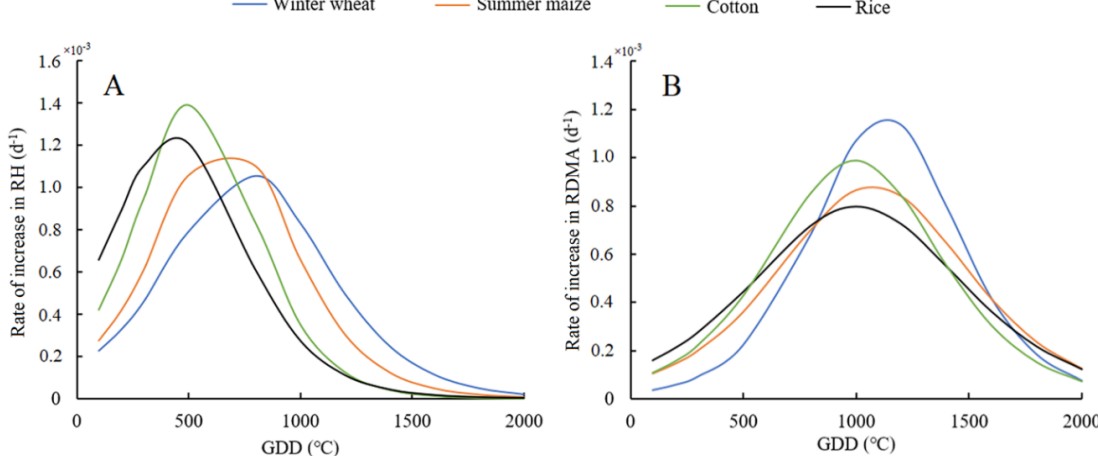

**Figure 3.** Relationships for each crop between GDD and the rates of increase in (**A**) RH and (**B**) RDMA.

### 3.2. Characteristics of the Fully Relative Logistic Growth Model of the Crops

RGDD was used to analyse the trends at different stages of crop growth. Equation (2) was used to simulate the growth of the four crops. The parameters of the fully relative logistic model are shown in Table 12, and the curves are plotted in Figure 4. RGDD at $LAI_{max}$ differed amongst the four crops in the order rice < summer maize < cotton < winter wheat (Figure 4A). When RGDD was regarded as a period of relative growth, LAI for the four crops peaked in the middle and late stages of growth—earliest for rice and latest for winter wheat. When RGDD was between 0 and 0.3, RLAI for winter wheat was 0, and RH was >0 and gradually increased, indicating the characteristic of "standing upright" after overwintering. RH and RDMA increased similarly amongst the four crops, but the slopes of the curves differ slightly (Figure 4B,C). RGDD for winter wheat, summer maize, and cotton at the same RH was in the order cotton < winter wheat < summer maize. RH in the same period of relative growth was largest for cotton and smallest for summer maize, and RH for rice was large in the early stage and small in the late stage. DMA at each stage of growth differed amongst the crops. By defining RGDD < 0.5 as the early period of growth and RGDD > 0.5 as the late period of growth, RDMA was in the order cotton > rice > summer maize > winter wheat during early growth and winter wheat > summer maize > cotton > rice during late growth. The crops with much early growth grew less in the later period, indicating that the rate of increase in DMA varied between the early and late stages of growth and varied amongst the crops.

**Table 12.** Parameters of the Fully Relative Logistic Model for the Indices of Crop Growth. RLAI-RGDD, RH-RGDD, and RDMA-RGDD are logistic models between RLAI and RGDD, RH and RGDD, and RDMA and RGDD, respectively. $a_f$, $b_f$, and $c_f$ are parameters of the fully relative logistic growth model. $c_f = 0$ in the relationship between RH-RGDD and RDMA-RGDD.

| Crop | RLAI-RGDD | | | RH-RGDD | | RDMA-RGDD | |
|---|---|---|---|---|---|---|---|
| | $a_f$ | $b_f$ | $c_f$ | $a_f$ | $b_f$ | $a_f$ | $b_f$ |
| Winter wheat | 18.010 | 53.41 | 34.99 | 3.283 | 8.499 | 4.493 | 8.099 |
| Summer maize | 7.385 | 28.15 | 20.32 | 3.192 | 8.013 | 4.143 | 7.028 |
| Rice | 6.380 | 26.23 | 20.18 | 2.222 | 6.469 | 3.525 | 5.829 |
| Cotton | 8.198 | 29.02 | 19.74 | 2.750 | 8.262 | 3.034 | 5.682 |

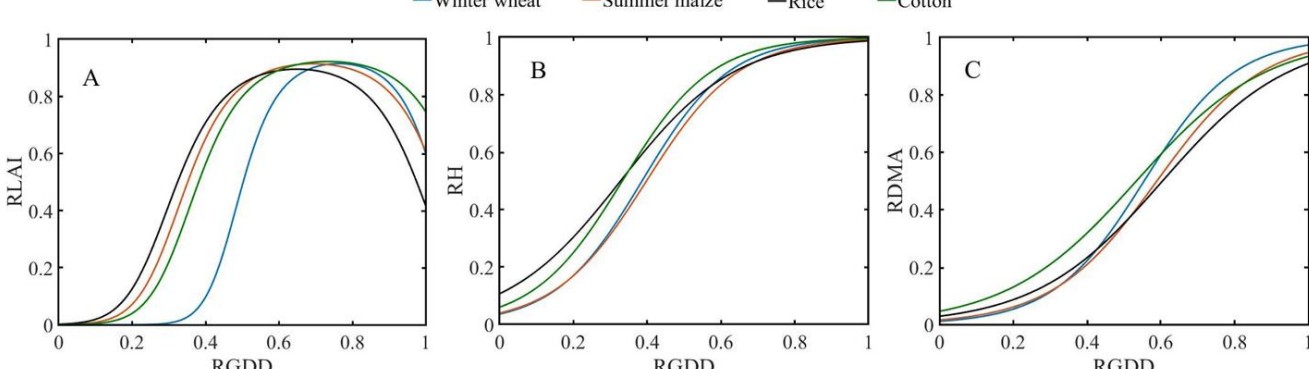

**Figure 4.** Fully relative logistic curves for each crop for (**A**) RLAI, (**B**) RH, and (**C**) RDMA.

Characteristic values of the fully relative logistic growth model were calculated to quantify the characteristics of growth for each crop (Table 13). When GDD was relative, the characteristic values of the models for the increases in RH and RDMA were very similar amongst the four crops. The fully relative logistic growth curves in Figure 4 were also very similar, indicating that the characteristics of RH and RDMA were similar for the four crops and that their properties of growth were essentially the same. The rate of increase in RH for each crop peaked at 1/3 of the period of growth, and the rate of increase in RDMA peaked between 1/2 and 4/5 of the period of growth. The stage of vigorous growth for each crop was about 1/3 of the entire period of growth, except for rice, which was about half of its entire period of growth. The periods of rapid increase in dry matter differed amongst the crops, being shortest for winter wheat and longest for cotton. The maximum rate of increase in RDMA was lowest for cotton and highest for winter wheat.

**Table 13.** Characteristic Values of the Fully Relative Logistic Model of the Crops. $RGDD_0$ is RGDD when RH or RDMA increased the fastest. $v_{max}$ is the maximum rate of increase in RH or RDMA. $RGDD_1$ is RGDD when RH or RDMA increased from slow to fast. $RGDD_2$ is RGDD when RH or RDMA increased from fast to slow. $RGDD_2 - RGDD_1$ is the relative length of the period of rapid increase in RH or RDMA.

|  | **Crop** | $RGDD_0$ | $v_{max}/d$ | $RGDD_1$ | $RGDD_2$ | $RGDD_2 - RGDD_1$ |
|---|---|---|---|---|---|---|
| RH-RGDD | Winter wheat | 0.39 | 2.12 | 0.23 | 0.54 | 0.31 |
|  | Summer maize | 0.40 | 2.00 | 0.23 | 0.56 | 0.33 |
|  | Rice | 0.34 | 1.62 | 0.14 | 0.55 | 0.41 |
|  | Cotton in Xinjiang | 0.33 | 2.07 | 0.17 | 0.49 | 0.32 |
| RDMA-RGDD | Winter wheat | 0.55 | 2.02 | 0.39 | 0.72 | 0.33 |
|  | Summer maize | 0.59 | 1.76 | 0.40 | 0.78 | 0.37 |
|  | Rice | 0.60 | 1.46 | 0.38 | 0.83 | 0.45 |
|  | Cotton | 0.53 | 1.42 | 0.30 | 0.77 | 0.46 |

The rates of growth in the same period of relative growth differed amongst the crops. The rates also differed for the same crop in different periods of growth. The relationships between the rates of increase in RH, RDMA, and RGDD are plotted in Figure 5. The rate of increase in RH for the four crops peaked in the early stage of growth (Figure 5A), and the rate of increase in RDMA peaked in the late stage of growth (Figure 5B). RH and RDMA for cotton were the first to reach $v_{max}$, which is consistent with the results in Table 11.

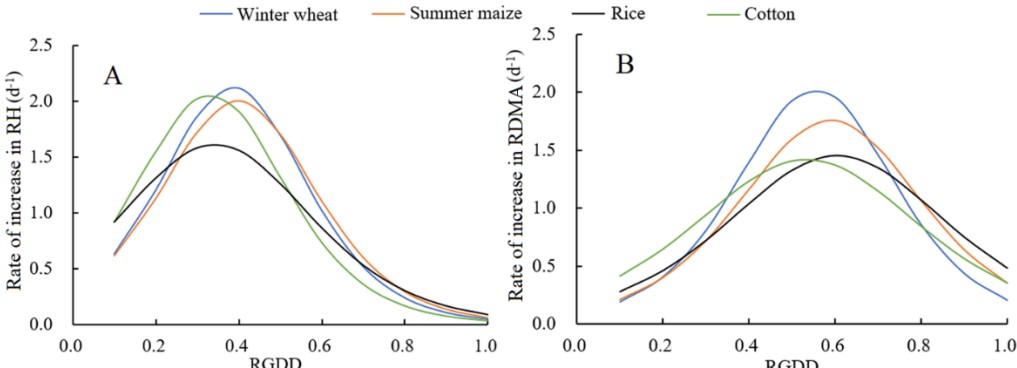

**Figure 5.** Relationships between relative growing degree days (RGDD) and the rates of increase in (**A**) RH and (**B**) RDMA for each crop.

### 3.3. Integrated Logistic Growth Model of the Crops

We calculated GDD and RGDD for winter wheat after the rising period to unify the fully relative logistic models of the four crops. The RLAI parameters of the modified fully relative logistic model of winter wheat were $a_f$ = 5.354, $b_f$ = 19.89, and $c_f$ = 13.08. The curves characterising the changes in RLAI for the four crops were then redrawn (Figure 6A). When the rising period was used as the starting point of the change of winter-wheat LAI, the order of increase in RLAI was similar amongst the four crops, but the order of decrease differed. We therefore only analysed increases. The curves of the modified fully relative logistic model when RGDD = 0–0.7 are shown in Figure 6B. The parameters of the fully relative models were averaged for the four crops to obtain a comprehensive fully relative logistic growth model (Equation (11)). The logistic curves are shown in Figure 7.

$$\begin{cases} \text{RLAI} = \dfrac{1}{1+e^{6.829-25.82\cdot\text{RGDD}+18.33\cdot\text{RGDD}^2}} \\ \text{RH} = \dfrac{1}{1+e^{2.862-7.811\cdot\text{RGDD}}} \\ \text{RDMA} = \dfrac{1}{1+e^{3.799-6.66\cdot\text{RGDD}}} \end{cases} \tag{11}$$

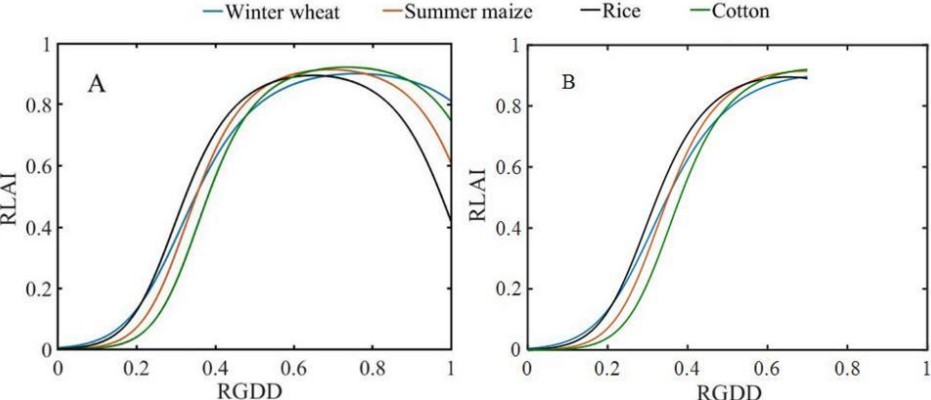

**Figure 6.** Modified fully relative logistic curves of the leaf area index (RLAI) for the crops for (**A**) increases and decreases and (**B**) increases. RGDD for winter wheat was calculated starting from the rising period.

Five sets of unmodelled data for each crop were used to evaluate Equation (11). Scatter plots between the measured and fitted values are shown in Figure 8. This comprehensive fully relative logistic model fitted the indicators well ($R^2$ > 0.75, RMSE < 0.15, RE < 5%) except for RLAI for summer maize and cotton (Figure 8D,J). Equation (11) can therefore describe the growth of the crops well.

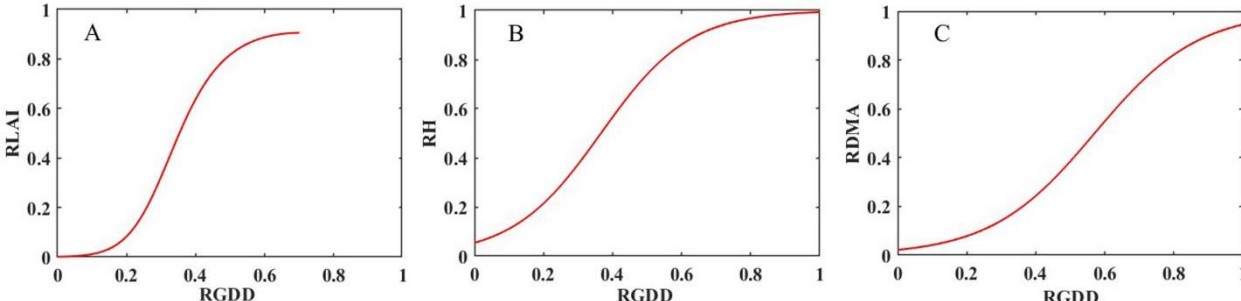

**Figure 7.** Comprehensive and fully relative logistic curves of the indices of growth for the four crops between (**A**) RLAI and RGDD, (**B**) RH and RGDD, and (**C**) RDMA and RGDD.

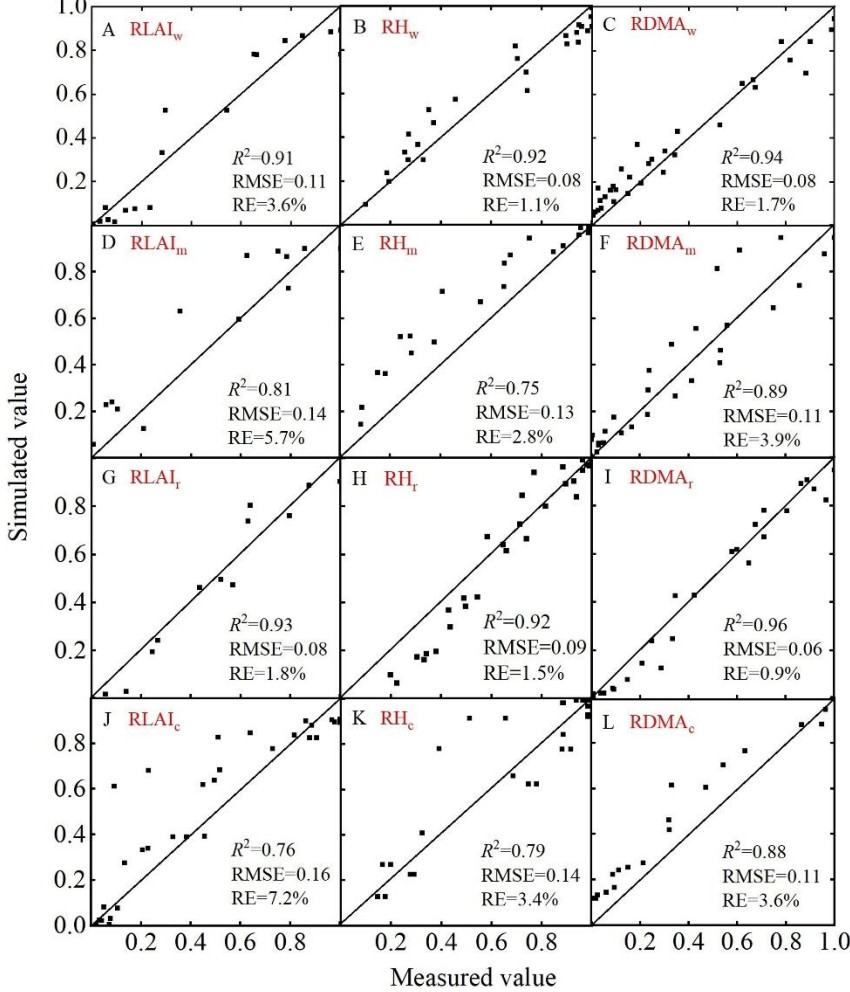

**Figure 8.** Validation diagram of the comprehensive and fully relative logistic growth model. $RLAI_w$, $RLAI_m$, $RLAI_r$, and $RLAI_c$ represent the leaf area indexes for winter wheat (**A**), summer maize (**D**), rice (**G**), and cotton (**J**), respectively. $RH_w$, $RH_m$, $RH_r$, and $RH_c$ represent plant heights for winter wheat (**B**), summer maize (**E**), rice (**H**), and cotton (**K**), respectively. $RDMA_w$, $RDMA_m$, $RDMA_r$, and $RDMA_c$ represent the accumulations of dry matter for winter wheat (**C**), summer maize (**F**), rice (**I**), and cotton (**L**), respectively.

### 3.4. Spatial Distribution of $LAI_{max}$ and $DMA_{max}$

Maps of the spatial distributions of $LAI_{max}$ and $DMA_{max}$ for analysing the spatial variability of each crop are shown in Figures 9 and 10. $LAI_{max}$ varied the most for winter

wheat, at 5.864, followed by summer maize, rice, and cotton. $LAI_{max}$ for winter wheat tended to increase and then decrease from northeast to southwest in the valleys of the Yellow and Huai Rivers and in the middle and lower reaches of the Yangtze River; gradually decreased from northeast to southwest in Xinjiang; and was highest in Henan, Shanxi, and Jiangsu Provinces (Figure 9A). The spatial variability of $LAI_{max}$ for summer maize gradually increased from northwest to southeast throughout China (Figure 9B). The spatial distribution of $LAI_{max}$ for rice had specific regional characteristics. $LAI_{max}$ for rice was low in northeastern, southwestern, and southern China and higher in the eastern coastal area (Figure 9C). The spatial variability of $LAI_{max}$ in Xinjiang was similar for cotton and winter wheat, large in the east and small in the west, and gradually increased from north to south (Figure 9D).

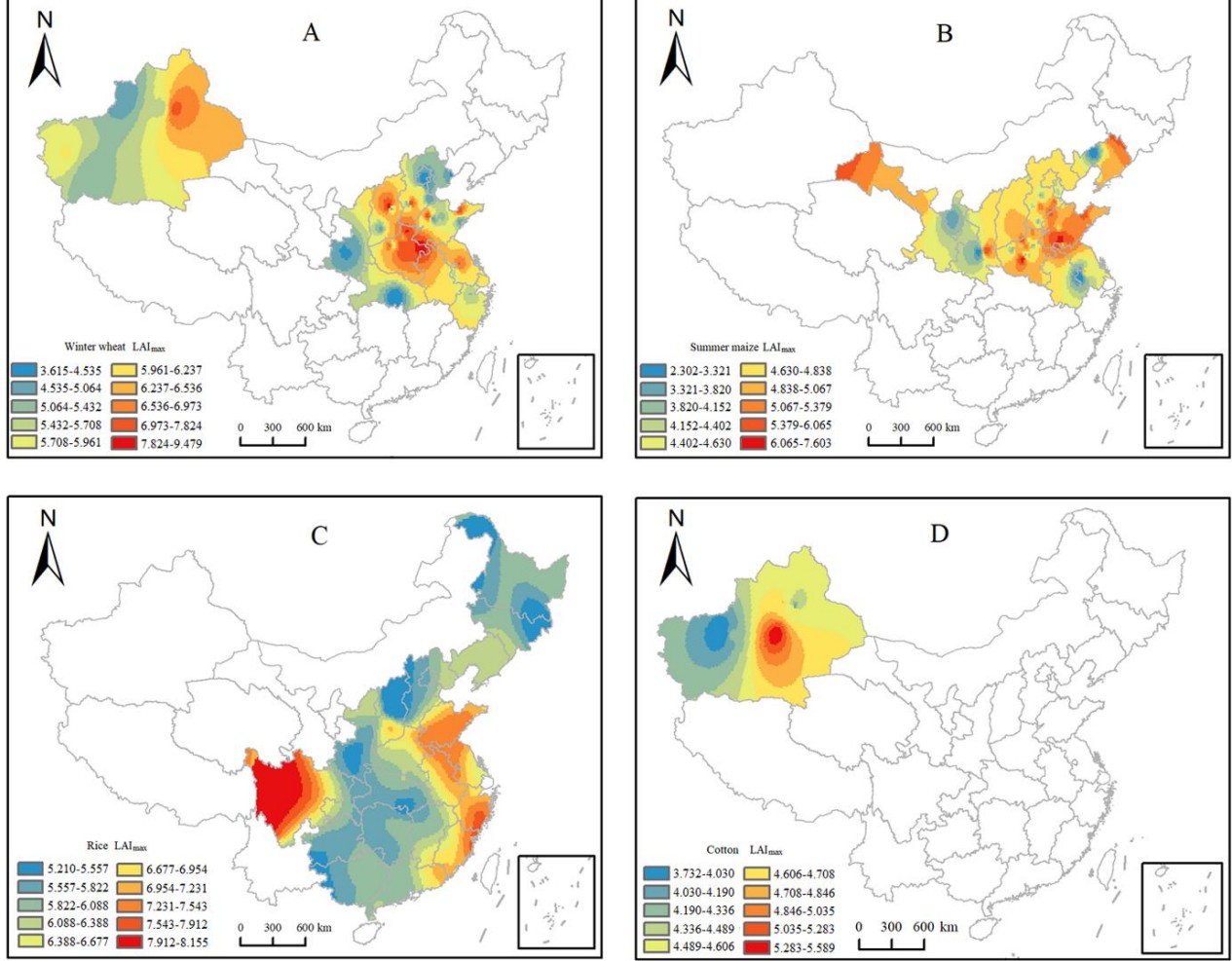

**Figure 9.** Spatial distribution of maximum leaf area index ($LAI_{max}$) for (**A**) winter wheat, (**B**) summer maize, (**C**) rice, and (**D**) cotton.

$DMA_{max}$ for winter wheat was large in southern Xinjiang and the middle and lower reaches of the Yangtze River and small in northern Xinjiang (Figure 10A). $DMA_{max}$ for summer maize was uniformly distributed, tending to increase and then decrease from northeast to southwest, and was largest in northwestern Gansu Province (Figure 10B). The spatial variability of rice $DMA_{max}$ was similar to that of $LAI_{max}$: low in southweatern and southern China and high in northeastern China (Figure 10C). The spatial distribution of cotton $DMA_{max}$ gradually increased from northwest to southeast in Xinjiang—the opposite trend to that of the distribution of $LAI_{max}$ (Figure 10D).

Statistics for $LAI_{max}$ and $DMA_{max}$ for the crops are presented in Table 14. The co-efficients of variation (CVs) for $LAI_{max}$ and $DMA_{max}$ for the four crops were all > 0.20. The CV for $LAI_{max}$ was in the order summer maize > winter wheat > rice > cotton. The CV for $DMA_{max}$ was in the order cotton > summer maize > winter wheat = rice. The results of Kolmogorov–Smirnov (K–S) tests for each index of crop growth were > 0.05 and normally distributed.

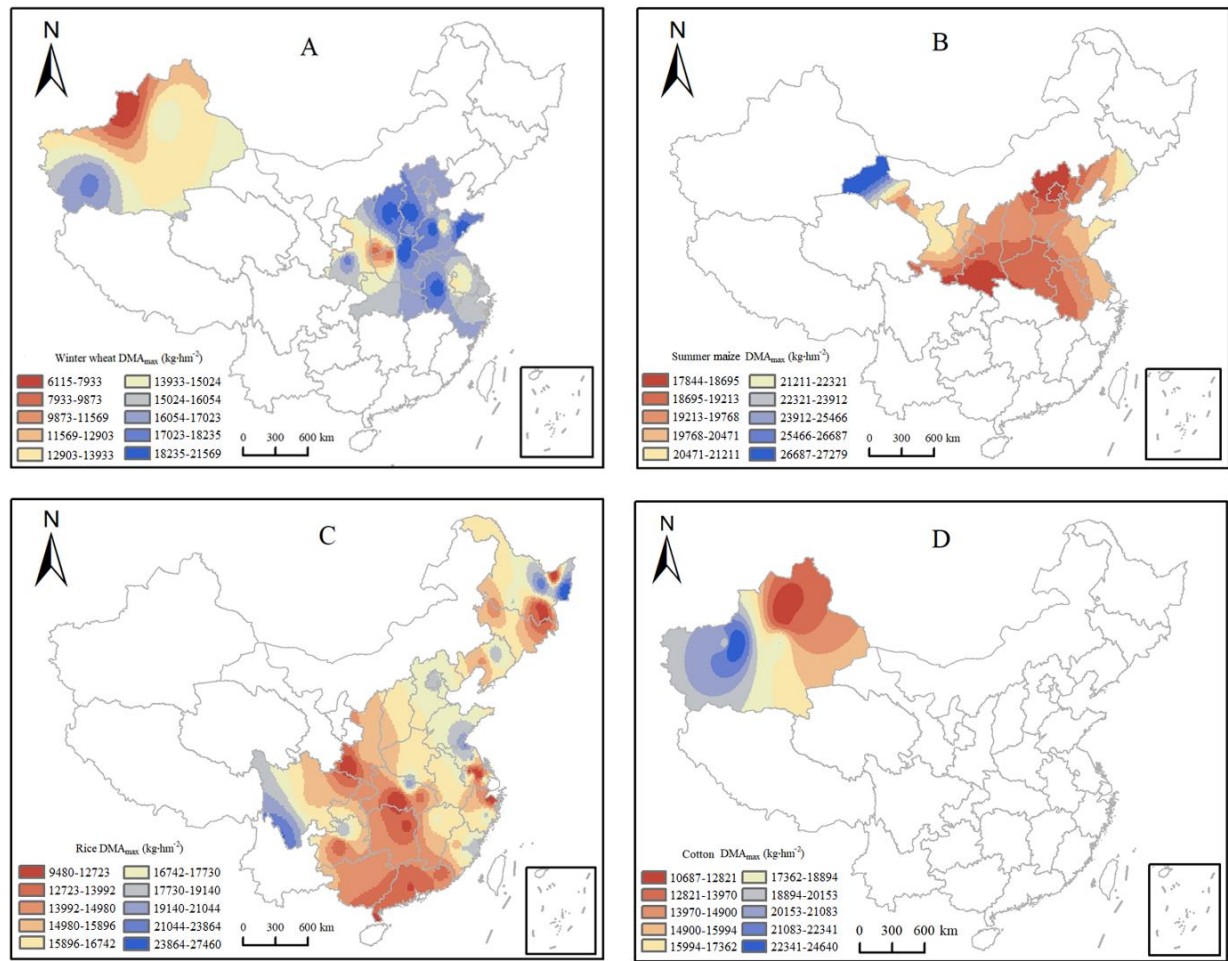

**Figure 10.** Spatial distribution of maximum dry-matter accumulation ($DMA_{max}$) for (**A**) winter wheat, (**B**) summer maize, (**C**) rice, and (**D**) cotton.

**Table 14.** Statistics for the Spatial Variability of the Indices of Crop Growth. CV, Coefficient Of Variation; K–S, Results of the K–S Test.

| Index | Crop | Mean | Standard Deviation | Minimum | Maximum | CV | K–S |
|---|---|---|---|---|---|---|---|
| | Winter wheat | 5.91 | 1.45 | 3.53 | 9.55 | 0.25 | 0.19 |
| $LAI_{max}$ | Summer maize | 5.01 | 1.40 | 2.26 | 12.22 | 0.28 | 0.06 |
| | Rice | 6.40 | 1.29 | 3.66 | 10.12 | 0.20 | 0.20 |
| | Cotton | 4.52 | 0.85 | 3.47 | 5.81 | 0.19 | 0.20 |
| | Winter wheat | 15,794.10 | 4159.44 | 6109.35 | 21,696.99 | 0.26 | 0.20 |
| $DMA_{max}$ | Summer maize | 19,826.32 | 5383.91 | 6976.74 | 27,279.07 | 0.27 | 0.19 |
| $(kg \cdot hm^{-2})$ | Rice | 16,061.03 | 4168.45 | 8945.50 | 27,502.31 | 0.26 | 0.20 |
| | Cotton | 16,807.65 | 5882.23 | 8547.97 | 25,754.73 | 0.35 | 0.20 |

### 3.5. Relationships between LAI_max, DMA_max, and Meteorological Factors

We established a binary quadratic relationship to determine the effects of meteorological conditions on $LAI_{max}$ and $DMA_{max}$, with W and $GDD_{max}$ corresponding to each $LAI_{max}$ and $DMA_{max}$ as independent variables, and $LAI_{max}$ and $DMA_{max}$ for each crop as dependent variables:

$$f(W, \text{GDD}_{max}) = m_1 W + m_2 \text{GDD}_{max} + m_3 W \cdot \text{GDD}_{max} + m_4 W^2 + m_5 \text{GDD}_{max}{}^2 + m_6 \quad (12)$$

where $f(W, \text{GDD}_{max})$ is $LAI_{max}$ or $DMA_{max}$, W is water consumption (mm) by the crop, $\text{GDD}_{max}$ is the maximum growing degree days of the crop (°C), and $m_1, \ldots,$ and $m_6$ are parameters. Ten data sets were randomly selected for each indicator to verify the equations. The parameters are presented in Table 15. The characteristics of the parameters differed, and the verification results were good. RE was < 10%, and $R^2$ was > 0.7 ($p < 0.01$).

**Table 15.** Parameters Between Meteorological Factors and the Maximum Values of the Indices of Crop Growth.

| Index | Crop | Parameter | | | | | | Validation Results | |
|---|---|---|---|---|---|---|---|---|---|
| | | $m_1$ | $m_2$ | $m_3$ | $m_4$ | $m_5$ | $m_6$ | $R^2$ | RE |
| LAI_max | Winter wheat | 0.028 | 0.029 | $-5.92 \times 10^{-6}$ | $-1.55 \times 10^{-5}$ | $-6.61 \times 10^{-6}$ | $-28.42$ | 0.82 | 6.6% |
| | Summer maize | $-0.067$ | $-0.014$ | $3.58 \times 10^{-5}$ | $-2.10 \times 10^{-6}$ | $1.80 \times 10^{-6}$ | 24.59 | 0.84 | 7.7% |
| | Rice | 0.023 | 0.022 | $-3.93 \times 10^{-6}$ | $-1.42 \times 10^{-5}$ | $-5.67 \times 10^{-6}$ | $-17.75$ | 0.74 | 6.5% |
| | Cotton | 0.078 | $-0.128$ | $2.05 \times 10^{-4}$ | $3.60 \times 10^{-4}$ | $1.28 \times 10^{-5}$ | 66.14 | 0.87 | 4.8% |
| DMA_max | Winter wheat | 378.26 | 176.23 | $-0.151$ | $-0.076$ | $-0.023$ | $-2.65 \times 10^5$ | 0.84 | 5.8% |
| | Summer maize | $-156.54$ | $-42.64$ | 0.078 | 0.013 | 0.007 | $7.11 \times 10^4$ | 0.88 | 5.6% |
| | Rice | 31.80 | 68.96 | 0.018 | $-0.047$ | $-0.033$ | $-3.09 \times 10^4$ | 0.93 | 4.2% |
| | Cotton | 2225.20 | $-449.99$ | $-0.570$ | $-1.553$ | 0.265 | $-2.02 \times 10^5$ | 0.85 | 4.7% |

## 4. Discussion

### 4.1. Comparison of the Semirelative and Fully Relative Logistic Growth Models of the Crops

This study found that both the semirelative and fully relative logistic models could describe the growth of the crops well, but the fully relative model could represent the mechanism of crop growth and determine the essential characteristics of growth of the different crop more intuitively than could the semirelative model. The GDD demand to reach maximum RLAI in the semirelative model was lower for cotton than summer maize [40,41]. RGDD in the fully relative model was higher for cotton than summer maize when RLAI peaked, indicating that summer maize needed more GDD in a short period of time due to the time needed for the growth of these two crops. Cotton is generally sown from April to May and harvested in September, and the average temperature demand throughout growth is 25 °C. The temperature was low and temperature accumulation was small in Xinjiang in the early stage of cotton growth, however, leading to slow cotton growth [42]. Summer maize is sown in June and harvested in late September, the growing season is short, and the average daily temperature is high, about 26 °C. GDD for summer maize therefore increased considerably in the short term, which is consistent with the model output. Similarly, RH in the semirelative logistic model was larger for summer maize than winter wheat at the same GDD, and H in the fully relative model was larger for winter wheat than summer maize at the same RGDD, indicating that the GDD demand for the increase in RH was higher for winter wheat than summer maize, which is consistent with the results in Table 10. Tables 10 and 11 indicate the demands of GDD and RGDD for the vigorous growth of typical crop RH and RDMA, respectively. Figures 3 and 5 show the trends of the rates of increase for the indicators of crop growth with GDD and RGDD separately.

In this study, crop growth indicators, GDD, and relative growth stage were linked; crops, meteorology, and models were comprehensively analysed; and the growth laws of

typical crops were compared, which has theoretical value for selecting suitable crops based on regional meteorological conditions, selecting sowing times, and quantitatively analysing the status of crop growth.

### 4.2. Comprehensive and Fully Relative Logistic Growth Model

The model parameters for RLAI and RGDD for summer maize, rice, and cotton were similar in the fully relative logistic growth models of the crops, and the parameters of the RH and RDMA models of the four crops were all similar (Table 12). Our study thus described the growth of the four crops using a unified and comprehensive fully relative logistic growth model and investigated the characteristics of change. Winter wheat had slightly different growth characteristics than those of the other three crops because it overwinters [43]. Winter wheat enters the overwintering period after tillering, at which stage the wheat almost stops growing and the leaves are all near the ground to ensure safe overwintering. The winter wheat then enters the bolting stage, when the plants gradually stand upright [44]. H therefore increases rapidly and LAI varies little during this period. These features are consistent with the semirelative and fully relative logistic curves. We calculated GDD and RGDD for winter wheat after this period to establish a unified logistic model of the four crops.

The increase in LAI was consistent for the four crops in the curves characterising the changes in RLAI (Figure 6A) when RGDD was between 0 and 0.7, and LAI tended to decrease when RGDD was >0.7. The different crops, however, had different rates of decrease at the same RGDD, in the order rice > summer maize > cotton > winter wheat. RLAI in the comprehensive fully relative logistic model (Equation (11)) could therefore only be fitted when RGDD = 0–0.7. The verification results of RLAI for cotton and summer maize both had REs > 5%. The parameters of the fully relative model of these two crops were compared with those of the comprehensive model; the differences of the parameters between the fully relative and comprehensive models were larger for cotton and summer maize than for winter wheat and rice, so the verification results were also poor. The comprehensive relative logistic model (Equation (11)) could nonetheless generally simulate the growth of the crops well, and the error was within an allowable range.

The area we used to gather data for the model nearly covered the main planting regions of the four crops, and the data also met the general requirements of the regional empirical model. The comprehensive fully relative logistic model could thus be used to simulate crop growth in different locations, climates, and soil conditions. The verification data also strongly influenced the utility of the model. We studied four crops, each with three growth indicators, and five sets of unmodelled data were randomly selected to verify the model for each growth indicator (for a total of 60 data sets). The model should thus be verified in the future using more comprehensive data.

### 4.3. Spatial Variability of $LAI_{max}$ and $DMA_{max}$ for the Crops

The maximum growth index in the integrated fully relative logistic growth model (Equation (11)) (e.g., $LAI_{max}$ or $DMA_{max}$) will directly affect *Ry*. Therefore, the maximum growth index was also crucial to the ability of the model to accurately describe crop growth [22]. LAI can represent photosynthetic capacity, which affects crop DMA [45,46], and yield is a part of DMA. We therefore studied the spatial distributions of $LAI_{max}$ and $DMA_{max}$ for the four crops, drew maps of the spatial distributions using ArcMap, with the planting area of each crop as the boundaries, and analysed the variability of each crop index. The results indicated that the distributions of $LAI_{max}$ and $DMA_{max}$ for each crop were closely associated with their growth characteristics and the climatic conditions in different regions, and had obvious regional features. For example, suitable soil water content during the growth of winter wheat is 60–80% [47], and its biological upper and lower limit temperatures are 0 and 32 °C, respectively [48]. The climate in Xinjiang, however, is dry, and soil water content is low [49,50]. The temperature in Xinjiang after the overwintering

period of wheat often fails to meet the demands for growth, leading to low DMAs in the region. The statistical results also indicated that each index was highly variable.

We performed a correlation analysis to identify correlations between $LAI_{max}$ and $DMA_{max}$ for each crop. The correlation coefficient, $r$, for the two rice indices was 0.405 ($p < 0.01$), but the indices for the other three crops were not correlated, indicating that the relationship between $LAI_{max}$ and $DMA_{max}$ depended on the physiological characteristics of the crops. We did not consider the differences between varieties when analysing the spatial variability of the crops, which may have affected the interpolation results. Future studies should thus focus on the differences in the distribution of growth indicators for different varieties of the same crop.

The spatial variability of the indicators of crop growth is critical for predicting yield [51,52]. We studied the spatial distributions of $LAI_{max}$ and $DMA_{max}$ for four typical crops, which will play a guiding role in the simulation of crop growth, the prediction of yield, regional agricultural planning, and the development of regional systems for planting crops.

### 4.4. Hydrothermal Coupling of LAI_max and DMA_max for the Crops

The maximum values of indices of crop growth are affected by many factors, such as soil water content, fertilisation, air temperature, heat requirements, and field management [53,54]. Water consumption and GDD throughout crop growth can indicate the demands of a crop for water and heat [55,56]. We used $GDD_{max}$ and water consumption as independent variables to study the variations in $LAI_{max}$ and $DMA_{max}$. Crop and meteorology were linked to ascertain the characteristics of the indicators of crop growth under different meteorological conditions. The results indicated that the maximum values of the indices of crop growth, water consumption, and $GDD_{max}$ were strongly correlated ($p < 0.01$).

Soil water content and temperature have theoretically specific thresholds for crop growth [57,58] (Stewart and Rattan, 2018; Ballesteros et al., 2018). The more vigorously a crop grows within a specific range of water consumption or $GDD_{max}$, the higher the maximum value of an index. $LAI_{max}$ and $DMA_{max}$ should tend to first increase and then decrease as W or $GDD_{max}$ increases, so the coefficients $m_3$, $m_4$, and $m_5$ in Equation (12) should all be negative. Except for $LAI_{max}$ for winter wheat and rice and $DMA_{max}$ for winter wheat in Table 10, however, the other parameters were not consistent with this rule, because crop growth is inseparable from factors such as basic soil fertility, the amount of fertilisation, and measures of field management [59,60], in addition to the influences of meteorological conditions such as humidity and heat. $LAI_{max}$ and $DMA_{max}$ also differed amongst the varieties of the same crop, which would affect the fitted results. Our study covered a wide area, and the CVs for $LAI_{max}$ and $DMA_{max}$ were large for the same crop, which also affected the final fitted result.

The status of crop growth depends on the variety [61] (Wannasek et al., 2019), soil fertility in different regions [62], and measures of field management [63]. Soil fertility varies less in small than large areas, and field cultivation and the management of water and fertilisation are more similar in small areas [64,65]. Studies in small areas should therefore focus on the relationships between the maximum indices of crop growth and meteorological conditions and analyse the variations in the maximum indices for different varieties with meteorological factors for accurately predicting crop growth and crop yield.

## 5. Conclusions

We compared logistic growth models of four typical crops, winter wheat, summer maize, rice, and cotton, following the results of previous studies, analysed the spatial distribution of LAI and DMA for the four crops, and established a binary quadratic coupling equation between $LAI_{max}$ and $DMA_{max}$ and two meteorological factors (W and $GDD_{max}$). The proposed comprehensive models provide a theoretical basis for analysing the growth status and predicting the yields of winter wheat, summer maize, rice, and cotton in China. We drew the following conclusions.

(1) The demand for GDD for the four crops in the semirelative logistic growth model when RLAI was highest was in the order rice < cotton < winter wheat < summer maize. H for the crops at the same GDD was in the order cotton > summer maize > winter wheat. The increase in DMA for the crops related to different GDD demands at different stages of growth. RGDD for the four crops in the fully relative logistic growth model when RLAI was highest was in the order rice < summer maize < cotton < winter wheat. RH at the same RGDD for the crops except rice was in the order cotton > winter wheat > summer maize, and the maximum rate of increase in RH was in the order winter wheat > cotton > summer maize. The order of RDMA differed between early and late crop growth.

(2) Both the semirelative logistic model and the fully relative logistic model could well simulate the changes in each indicator of crop growth. The fully relative logistic model could intuitively represent the growth characteristics of the crops better than the semirelative logistic model. We established a comprehensive logistic model that could describe the growth of the four crops, and the verification results were good.

(3) The spatial distributions of $LAI_{max}$ and $DMA_{max}$ for the four crops were highly variable, and the variations and levels of $LAI_{max}$ and $DMA_{max}$ differed amongst the crops. Water consumption and $GDD_{max}$ simulated crop $LAI_{max}$ and $DMA_{max}$ well.

**Author Contributions:** Conceptualization, F.L. and Q.W.; methodology, L.S.; software, L.S. and Y.L.; validation, Q.W., L.S. and Y.L.; formal analysis, F.L.; investigation, Y.L.; resources, M.D.; data curation, Y.L.; writing—original draft preparation, F.L.; writing—review and editing, Y.L., W.T. and L.S.; supervision, Q.W. and M.D.; project administration, Q.W.; funding acquisition, M.D., W.T. and L.S. All authors have read and agreed to the published version of the manuscript.

**Funding:** This work was supported by National Natural Science Foundation of China (51979220, 52109064, 52179042), the Major Science and Technology Projects of the XPCC (2021AA003–2), and Natural Science Basic Research Plan in Shaanxi Province of China (No.2021JM-320).

**Institutional Review Board Statement:** Not applicable.

**Informed Consent Statement:** Not applicable.

**Conflicts of Interest:** The authors declare no conflict of interest.

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
