# Peer review of "Integrated Growth Model of Typical Crops in China with Regional Parameters"

_water, doi:10.3390/w14071139_

Round 1

Reviewer 1 Report

In the attached.

Author Response

Dear reviewer,

Thank you for your positive comments.

Best wishes,

Lijun Su

Reviewer 2 Report

The article is very extensive. Therefore, I propose to select 1 maximum of 2 crops for future research and to reduce the area of observation.

Reviewer 3 Report

Comments

SUMMARY

The paper addresses the research area related to agriculture in the MDPI Water journal. I believe that the target journal is an appropriate forum for this article. The objectives of this study were to (i) compare the characteristics of the growth of four crops based on the logistic and (ii) clarify the characteristics of spatial distribution and hydrothermal coupling for the indices of maximum crop growth in the logistic model.

BROAD COMMENT

This study is of great importance for agriculture in China. The Introduction section is written with recent references. The methods were well described and in detail allowing a good understanding of the results of the study. They discussed well the results of the study. However, I have some concerns about the different parts of the manuscript. I suggest a minor revision to address a few issues. If the authors address carefully the comments, I’ll recommend publication of the manuscript in the journal.

SPECIFIC COMMENTS

  • Table 2 and Table 3a: Which soil classification system are you using to characterize the soils?
  • What are the implications of the findings of the paper for agriculture in China?

Please, include them in the conclusion and abstract sections.

  • Line 347: Equation 12 seems to have some missing variables.
  • Figure 6: To make the comparison easier, harmonize the x-axis to 0-1 on both graphs a and b.
  • Figure 7: To make the comparison easier, harmonize the x-axis to 0-1 on all the graphs.
